



**Cloud properties over the Southern Ocean during the MARCUS field**
**campaign**
Baike Xi[1], Xiquan Dong[1], Xiaojian Zheng[1], and Peng Wu[2]
[1]Department of Hydrology and Atmospheric Sciences, University of Arizona, Tucson, AZ, USA
[2]Pacific Northwest National Laboratory, Richland, WA, USA
**Correspondence:** Baike Xi (baikex@arizona.edu)
**Abstract.** To investigate the cloud properties over the Southern Ocean (SO), the MARCUS field
campaign (41 to 69 °S; 60 to 160 °E) was conducted from October 2017 to March 2018, using
ship-based measurements. To examine cloud properties over the mid-latitude and Polar regions,
the study domain is separated into northern (NSO) and southern (SSO) parts of the SO with a
demarcation line of 60 °S. The total cloud fractions (*CF*s) were 77.9 %, 67.6 %, and 90.3 % for
the entire domain, NSO and SSO, respectively, indicating that higher *CF*s were observed in the
Polar region. Low-level clouds, deep cumulus, and shallow cumulus clouds are the three most
common cloud types over the SO. For single-layered low-level clouds, mixed-phase clouds
dominate with an occurrence frequency (*Freq*) of 54.5 %, while the *Freq* of the liquid and ice
clouds were 10.1 % (most drizzling) and 17.4 % (least drizzling). The meridional distributions of
low-level cloud boundaries are nearly independent of latitude, whereas the cloud temperatures
increased ~ 8 K and atmospheric precipitable water vapor increased from ~5 mm at 69 °S to ~18
mm at 43 °S. The mean cloud liquid water paths over NSO were much larger than those over SSO.



Most liquid clouds occurred over NSO with very few over SSO, whereas more mixed-phase clouds
occurred over SSO than over NSO. There were no significant differences for ice cloud *Freq*
between NSO and SSO. These results will be valuable for advancing our understanding of the
meridional and vertical distributions of clouds and can be used to improve model simulations over
the SO.
**1.  Introduction**
The Southern Ocean (SO) is one of the cloudiest and stormiest regions on the Earth (Chubb et
al., 2013). The majority of the aerosols are naturally produced via oceanic sources given the remote
environment. However, we have limited knowledge about cloud formation processes within such
clean environments and their associated aerosol and cloud properties. The unique nature of the SO
region features low-level supercooled liquid and mixed-phase clouds, which is significantly
different from the subtropical marine boundary layer (MBL) clouds where warm liquid clouds are
dominant (Dong et al., 2014; Wu et al., 2020; Zhao et al., 2020). Large biases in cloud amount and
microphysics over the SO in the Coupled Model Intercomparison Project phase 5 (CMIP5) climate
models result in a near 30 W m$^{-2}$ shortwave radiation deficit at the top of the atmosphere (TOA)
(Marchand et al., 2014; Stanfield et al., 2014, 2015), which further leads to unrealistic cloud
feedbacks and equilibrium climate sensitivity (Bony et al., 2015; Stocker et al., 2013). Meanwhile,
the efficiency of aerosol-cloud interaction (ACI) over the SO was found to be crucial for the
models' sensitivities to the radiation budget. A new aerosol scheme in the Hadley Centre Global
Environmental model can dampen the ACI and suppress negative clear-sky shortwave feedback,
both of which contribute to a larger climate sensitivity (Bodas-Salcedo et a., 2019).



A climate sensitivity study using CMIP6 general circulation models (GCMs) shows much
higher temperature variations across 27 GCMs in response to doubled $CO_2$ than those in CMIP5,
which may have resulted from the decreased extratropical low-level cloud cover and cloud albedo
over the SO in CMIP6 (Zelinka et al., 2020). Low-level clouds are a key climate uncertainty and
can explain 50 % of the inter-model variations (Klein et al., 2017) because conversion from liquid
cloud droplets to ice cloud particles decreases the cloud albedo and reduces the reflected shortwave
radiation at TOA. Models, however, have difficulties accurately partitioning the cloud phase
(Kalesse et al., 2016). The phase changes in mixed-phase clouds over the Arctic have proved to
affect the cloud lifetime and radiative properties significantly, that is, converting from ice cloud
particles to liquid cloud droplets may increase the cloud optical depth and the reflected shortwave
radiation at TOA (Morrison et al., 2012). In contrast, models that allow mixed-phase clouds to
glaciate rapidly can produce 30% more warming from doubling $CO_2$ (McCoy et al., 2014).
Phase transition processes have been investigated by several groups using both satellite and
ground-based measurements. A study (Mace and Protat, 2018) found that there are more mixed-
phase clouds over the SO measured from the ship than retrieved from CloudSat and CALIPSO
measurements because the satellites cannot accurately measure clouds below ~1 km. A previous
study (Lang et al., 2018) used a model to investigate the clouds under post cold frontal systems
and found large biases in model simulations and concluded that the cloud cover and radiative
biases over the SO are highly regime dependent. Of all cloud types, low-level clouds are primarily
responsible for the biases in the model simulations due to the lack of reliable measurements, which
leads to a poor understanding of the conditions where these clouds form and the phase(s) that result.
In other words, a physical representation of clouds, especially for low-level clouds, is unclear but
truly necessary for improving model simulations. Therefore, reliable observations of the cloud





macro- and micro-physical properties from ground-based active and passive remote sensors are
crucial for the improvement of model simulations.

Previous studies show that cloud phase is primarily dependent on cloud temperature, and the

transition from one cloud phase to another will modify the cloud optical properties, which further
affects the radiation budgets (Hu et al., 2010; Intrieri et al., 2002; Morrison et al., 2012). Based on
satellite observations and retrievals, they found that supercooled liquid water (SLW) clouds are
most common in the low-level clouds over the SO, where 80% of low-level clouds contain SLW
in a wide range of cloud temperatures from 0°C to −40°C (Hu et al., 2010). The formation of SLW
clouds is usually related to strong boundary layer convection. However, when ice nuclei exist in
the mixed-phase clouds, the ice particles can grow quickly and become bigger through consuming
supercooled liquid water drops. The SLW is inherently unstable due to the higher vapor pressure
over liquid than over ice and the quicker vapor deposition on ice particles than on liquid droplets
(Intrieri et al., 2002). As the supercooled liquid cloud droplets glaciate to ice particles, the cloud
layer becomes darker because the ice particles scatter less shortwave radiation and absorb more
radiation in the near IR wavelength regime. It is unclear, however, what role these ice particles
play in the low-level clouds over the SO, which includes the impact on drizzle development.
During HIAPER Pole-to-Pole Observation (HIPPO) campaigns, the study in Chubb et al. (2013)
found that there are rarely ice particles in non-drizzling and light drizzling clouds over the SO,
which may imply that the ice particles in the mixed-phase clouds may modulate the drizzle
formation.

To investigate the aerosol and cloud properties over the SO, a field campaign called the

Measurements of Aerosols, Radiation, and Clouds over the Southern Ocean (MARCUS) was
conducted using the ship-based measurements between Hobart, Australia, and the Antarctic during





the period October 2017-March 2018. The Department of Energy (DOE) Atmospheric Radiation
Measurement (ARM) Mobile Facility (AMF2) was installed on the Australian icebreaker *Aurora*
*Australis,* which voyaged from Hobart, Tasmania to the Australian Antarctic stations of Casey,
Mawson, and Davis, as well as Macquarie Island as illustrated in Fig. 1. Another field campaign,
called South Ocean Clouds, Radiation, Aerosol Transport Experimental Study (SOCRATES) field
campaign was conducted during austral summer from January 15 to February 26, 2018, which in-
situ measurements may use as a reference for this analysis. The SOCRATES domain has shown
as black dotted rectangle box in Fig. 1. The objectives of the MARCUS campaign are to investigate
the vertical distribution of boundary layer clouds and reveal the reasons why the mixed-phase
clouds are common in the warm season (McFarquhar et al., 2016), which will be our focuses for
this study.
MARCUS ship-based instruments include AMF2 cloud radar, lidar, microwave radiometer,
micropulse lidar, radiosonde sounding, precision solar pyranometer and precision infrared
radiometer, as well as aerosol sensors. Through these comprehensive observations over the SO,
we are tentatively answering the following three scientific questions:
(1)    What is the total cloud fraction over the SO during MARCUS, as well as vertical
and meridional variations in cloud fraction?
(2)    What are the dominant cloud types over the SO, their associated cloud properties,
as well as their vertical and meridional distributions?
(3)    What are the vertical and meridional distributions of the low-level clouds over the
SO?





This manuscript is organized as follows: the data and method are introduced in section 2. The
statistical results for all clouds during MARCUS are summarized in section 3. The low-level cloud
properties are described in section 4, followed by a summary and conclusions in section 5.

**2. Data and Method**
The cloud properties analyzed in this study are derived from the data collected by AMF2,
including the 95-GHz W-band cloud radar, ceilometer, micropulse lidar, microwave radiometer,
aerosol observation system (AOS), meteorological measurements (MET, includes the following
data: temperature, pressure, specific humidity, wind direction, and speed) on the ship, rain gauge
and the radiosonde soundings. The combined cloud radar and ceilometer measurements can
provide the cloud boundaries as long as there are no optically thin clouds and the cloud-base
heights ($H_{base}$) are not greater than the upper limit (7.7 km) of the ceilometer. The micropulse lidar
will be used to identify optically thin clouds and the clouds with $H_{base} > 7.7$ km. A previous study
has shown that these additional clouds detected by the micropulse lidar can be a non-negligible
supplement to the total cloud fraction (Mace et al., 2021). There are about 4 to 5 radiosonde
soundings per day. We adopted a linear interpolation method based on these daily soundings to
achieve a better temporal resolution of temperature, pressure, and specific humidity. The method
considers MET measurements to ensure vertical continuity and adjacent soundings for temporal
continuity. Using these interpolated atmospheric profiles, cloud temperatures can be accurately
estimated.
The cloud occurrence frequency can be determined through two steps: the column cloud
fraction is simply the ratio of cloudy samples to the total observations in every 5-min; the
occurrence frequency for each type of cloud during the entire time period equals the ratio of the



number where column cloud fraction is greater than zero to the total 5-min samples. The cloud
liquid water path (*LWP*) and atmospheric precipitable water vapor (*PWV*) are retrieved based on a
physical-iterative algorithm using observations of the microwave radiometer brightness
temperature at 23.8 and 31.4 GHz with uncertainties ranging from 15 to 30 g m$^{-2}$ (Marchand et al.,
2003). It is important to note that the brightness temperature biases switch signs among different
climatological regions because a threshold of 5 ℃ in cloud-base temperature was used in their
physical retrievals. Therefore, we propose an extra step to determine the uncertainties during
MARCUS. Both the AOS and rain gauge measurements were used to determine whether rain is
reaching the surface qualitatively, but not quantitatively in this study. All the measurements were
averaged over 5 minutes except radar reflectivity, Doppler velocity, and spectrum width used in
Section 4.3. The detailed classification method will be introduced in Section 4.1. In brief, we used
the measurements of interpolated sounding, microwave radiometer retrieved *LWP*, radar
reflectivity, Doppler velocity and spectrum width to classify the cloud phase in each radar range
volume of low-level clouds during MARCUS. We also used ERA5 reanalysis data to study the
environmental conditions during MARCUS and calculated the lower tropospheric stability (LTS)
and estimated inversion strength (EIS) when the low-level clouds appeared along the shiptracks.
A classification method developed in Xi et al. (2010) was used to calculate the occurrence
frequencies of different types of clouds and their corresponding cloud macrophysical properties,
e.g., cloud base ($H_{\mathrm{base}}$) and top ($H_{\mathrm{top}}$) heights, cloud thickness ($\Delta H$), and *LWP*. The relative
contributions of mixed-phase, liquid and ice clouds to the single-layered low-level clouds as well
as their drizzling status are analyzed in this study. To further investigate the drizzling status under
different cloud phases, we also calculated their LTS and EIS. The latitudinal and longitudinal





variations of the single-layered low-level clouds as well as their vertical distributions are also
explored in this study.

**3. Statistical results for all clouds during MARCUS**
The occurrence frequencies of total cloud cover and different types of clouds and their
associated properties over the entire study domain during MARCUS are presented in Figs. 2-4. In
order to examine the cloud properties over the mid-latitude and Polar regions, we separate the SO
domain into northern (NSO, north of $60^{\circ}$S) and southern (SSO, south of $60^{\circ}$S) parts using a
demarcation line of 60ºS. A total of 2,447 hours cloud samples were collected during MARCUS
in this study, in which 1,181 hours of samples were located in the NSO and 1,266 hours of samples
were collected from the SSO. It is important to note that adding micropulse lidar measurements
increased the total samples of non-liquid-containing clouds by ~20% because micropulse lidar is
more sensitive to optically thin clouds than cloud radar. However, micropulse lidar signals are
usually attenuated and cannot provide a meaningful signal when the liquid cloud layer is thicker
than a couple of hundred meters (Sassen, 1991).
Figure 2 shows the vertical distributions of total cloud cover over the entire domain, as well as
over NSO and SSO. For the vertical distributions, the occurrence frequencies of total cloud
increase from the first radar gate (~ 226 m) to ~700 m, then monotonically decrease with altitude
with a few small increments at different levels, especially over SSO. We can draw the following
conclusions by comparing the occurrence frequencies of the total cloud between NSO and SSO. 1)
The SSO has more cloudiness than the NSO under 7 km, while the NSO has more cloudiness than
the SSO above 7 km. 2) Below 3 km, the occurrence frequencies of clouds over the NSO decrease
dramatically from 37 % at an altitude of ~700 m to 16 % at 3 km and from 45 % to 28 % over the





SSO, which is similar to the vertical distributions of the low-level clouds over some Northern
Hemisphere mid-latitude regions, such as Eastern North Atlantic (ENA, Dong et al., 2014). The
occurrence frequencies measured during MARCUS are much lower than these shown in Fig. 8 of
Mace et al. (2009) throughout the entire vertical column between the same range of latitudes,
especially, the occurrence frequencies during MARCUS are almost half of these measured by
CloudSat and CALIPSO from 1 to 3 km. The reason has been explained in Xi et al. (2010), that is,
a comparison of occurrence frequencies between measurements of two different platforms can
only be performed under an equivalent spatial-to-temporal resolution. In other words, our results
were calculated under 5-min temporal resolution, and the results in Mace et al. (2009) were
statistically in the 2 ° grid box. Therefore, the comparison between these two results is not
reasonable. To make a fair comparison, one has to know the cloud amount at each area or time
step, then the product of amount and frequency is independent of either temporal and spatial
measurement.

To compare with other studies, we calculated the cloud fractions ($CF$s) of total and different

types of clouds. The total $CF$s were 77.9 %, 67.6 %, and 90.3 % for the entire domain, NSO and
SSO, respectively, indicating that 22.7 % more clouds occurred in the Polar region than in the mid-
latitude region. The total $CF$ over the entire domain is very close to the 76 % calculated by Mace
and Protat (2018) using ship-based measurements during the Cloud, Aerosols, Precipitation,
Radiation and Atmospheric Composition (CAPRICORN) field experiment. The total $CF$ over the
SSO is very close to that estimated by using the complementarity of CALIOP lidar aboard
CALIPSO and CPR aboard CloudSat (DARDAR version 2 data) from Listowski et al. (2019).

Figure 3 shows the occurrence frequencies of categorized clouds and their cloud boundaries

using the maximum $H_{top}$ and the minimum $H_{base}$. The definition of each type of cloud follows the

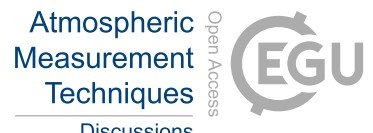

method of Xi et al. (2010). A brief description of the classification of cloud types is as follows.
The single-layered low-level clouds (LOW) is the fraction of time when low clouds with $H_{top} \leq 3$
km occur without clouds above them. Middle clouds (MID) range from 3 to 6 km without any
clouds below and above, while high clouds (HGH) have $H_{base} > 6$ km with no cloud underneath.
Other types of clouds are defined by different combinations of the above three types, middle over
low (MOL), high over low (HOL), high over mid (HOM), and the cloud column through the entire
troposphere is defined as HML. Three types, MOL, HOM, and HML, include both contiguous and
non-contiguous cloud layers, and their thicknesses may be overestimated when clear layer(s) are
present between any two cloud layers.

As illustrated in Fig. 3a, the single-layered low-level (LOW), deep cumulus or multi-layered

(HML), and shallow cumulus (MOL) clouds are the three dominant types of clouds over the SO.
Comparing the clouds between NSO and SSO, all types of clouds in SSO are higher than those in
NSO except HOL. The differences range from less than 1 % (LOW) to more than 10 % (MOL).
Comparing the clouds over mid-latitude oceans between the two hemispheres, i.e., between NSO
and ARM ENA site (Dong et al., 2014), we find: (1) The total cloud fractions (*CF*s) are close to
each other (67.6 % over NSO vs. 70.1 % at ARM ENA); (2) LOW *CF*s are 22.9 % vs. 27.1 %,
which is the dominant type of cloud in both regions; and (3) Both MOL and HML clouds, including
underneath low clouds, are 14.2 % and 16.5 % over NSO, much higher than those (4.2 % and
12.1 %) at ARM ENA site, indicating that there are more shallow and deep convective clouds over
NSO than over ENA.

Figure 3b shows the vertical locations of different types of cloud layers, which represent the

maximum $H_{top}$ and minimum $H_{base}$, as well as their deepest $\Delta H$ for any type of cloud. Nearly all
$H_{top}$ and $\Delta H$ values over NSO are higher or deeper than those over SSO, presumably due to stronger



solar radiation and stronger convection over NSO. $H_{base}$ values basically followed their cloud-top
counterparts with a couple of exceptions. These cloud properties are closely associated with large-
scale dynamic patterns and environmental conditions. By analyzing the ERA-I reanalysis (not
shown), the 850 hPa geopotential heights show persistent westerlies with slightly higher
geopotential heights over the northwest corner of the domain, which may closely relate to the
higher $H_{top}$ over NSO than over SSO. Furthermore, the boundary layer over NSO is relatively more
stable than over SSO based on lower troposphere stability analysis.

When we plot the probability density functions (PDFs) of cloud $LWP$s for different types of

clouds, we find that the PDFs of $LWP$s for HGH and HOM peak at less than 10 g m$^{-2}$. These results
make physical sense because HGH clouds should not contain any liquid droplets, and most HOM
clouds, especially those over SSO, should be ice phase dominant. In addition, the 10 g m$^{-2}$ of $LWP$
is close to the uncertainty of the $LWP$ retrieval in Marchand et al. (2003). Therefore, this value is
used as a threshold for all types of clouds, which leads to less than one percent reduction of the
total samples. As shown in Fig. 4a, the $LWP$s (> 10 g m$^{-2}$) for all types of clouds are much higher
over NSO than over SSO because the low-level and shallow convective clouds in the mid-latitudes
contain more liquid water than those in Polar regions. The mean $LWP$s for liquid containing low-
level and middle-level clouds over NSO, such as LOW, MID and HOL, range from ~130 to 150 g
m$^{-2}$, while the mean $LWP$s for shallow and deep convective clouds, such as MOL and HML, are
two times higher (~270 g m$^{-2}$) than the mean $LWP$ of LOW, MID and HOL. Note that the mean
$LWP$s for most types of clouds over the SSO are much lower than those over the NSO, except for
the LOW clouds.

Table 1 provides a summary of the average, standard derivation, minimum and maximum for

cloud boundaries, liquid water path and the percentage of multi-layered cloud for each cloud type



over the SO. Non-contiguous clouds over the SO occur very frequently, especially for HOM and
HML. The *LWP* for single-layered clouds is greater than that for multi-layered clouds. The *LWP*
for single-layered HML almost doubles that for multi-layered HML.

The occurrence frequencies of *LWP*s ($> 10$ g m$^{-2}$) over NSO and SSO contradict their cloud

*LWP* counterparts, as demonstrated in Fig. 4b. To further investigate the amount of available
precipitable water vapor (*PWV*), we found that mean *PWV* values in SSO are at least 2 to 3 times
less than those in NSO for the same types of clouds (figure not shown). Note that the samples of
MID, HGH, and HOM clouds are excluded from this study when they have *LWP*s less than 10 g
m$^{-2}$, since these low *LWP*s are within the retrieval uncertainty of cloud *LWP* and hence may not
contain any liquid cloud droplets. The higher *LWP*s, larger cloud droplets, drizzle drops and ice
particles, and greater drizzling occurrence frequencies over NSO (which is discussed later) will
lead to the quick dissipation of clouds over NSO. In contrast to NSO, the SSO cloud *LWP*s and
particle sizes are much smaller with fewer drizzling events, which increases cloud lifetime relative
to NSO. The 67.6 % and 90.3 % *CF*s over NSO and SSO provide strong evidence for this argument.
We can draw the following conclusions by comparing the cloud properties between NSO and SSO
in Figs. 3 and 4. The LOW fraction, thickness, and *LWP* over NSO and SSO are comparable to
each other. For other types of clouds, cloud thicknesses are similar to each other or slightly deeper
over NSO, but the cloud *LWP*s over NSO are much larger than those over SSO, resulting in more
precipitation events over NSO. As pointed out in (Albrecht 1989), more precipitation events may
reduce the cloud lifetime. This argument is consistent with the results shown in Figs. 2 and 3a for
all clouds except for HOL. Cloud lifetimes over NSO are shorter than those over SSO, which leads
to lower *CF*s over NSO than over SSO.





**4. Single-layered low-level clouds**
As discussed in Section 3, single-layered low-level clouds (LOW) are the dominant cloud type
in both northern (NSO) and southern (SSO) parts of the SO. Figs. 3 and 4 further reveal that LOW
cloud type is the only one having comparable *CF*, cloud, thickness, *LWP* over both NSO and SSO.
This warrants further study: Are the cloud phases, properties, and vertical and meridional
variations of LOW clouds over these two regions similar to each other or significantly different?
**4.1. Cloud phase**
In this study, cloud boundaries are determined by combining cloud radar, ceilometer and
micropulse lidar measurements at a temporal resolution of 5-min. The cloud phase, liquid water
droplets or ice particles, are determined in each radar range volume. A flow chart for classifying
the phases of single-layered low-level clouds is drawn in Fig. 5. The determination of warm liquid
clouds is straightforward using both cloud-base ($T_\mathrm{base}$) and -top ($T_\mathrm{top}$) temperatures greater than 0
°C, and cloud *LWP*s greater than the threshold (10 g m$^{-2}$). The determination of supercooled liquid
clouds is slightly complicated. When either $T_\mathrm{base}$ or $T_\mathrm{top}$ is below 0° C, and cloud *LWP*s are greater
than the threshold, the radar Doppler spectrum width (*WID*) and velocity ($V_\mathrm{d}$) are used for the
determination of supercooled liquid water clouds. If the majority (10 seconds of original radar
measurements) of *WID* are less than 0.4 m s$^{-1}$ and $V_\mathrm{d}$ are equal to or less than 0.0 m s$^{-1}$ (updrafts)
in the volume, then this range volume is defined as supercooled liquid clouds.
Mixed-phase clouds are determined when the medians (calculated from 10 seconds of original
radar measurements) of *WID* is greater than 0.4 m s$^{-1}$ and $V_\mathrm{d}$ is greater than 0.0 m s$^{-1}$ (downdrafts)
due to the existence of large ice particles in the clouds. If cloud *LWP* is below the threshold, then
it is defined as an ice cloud, otherwise, it is defined as a mixed-phase cloud. It is worth mentioning
that large ice particles, which grow through vapor deposition or rime processes, dominate the radar





reflectivity and are heavier than cloud droplets. Therefore, these large ice particles not only
broaden the spectrum width but also have relatively large fall speeds.

To further evaluate our classification method, we compared the classified mixed-phase and ice

clouds with the micropulse lidar linear depolarization ratios (*LDR*) as an extra measure. The *LDR*
ranges follow the method in Shupe et al. (2005), which are $0.11 < LDR < 0.15$ for mixed-phase
clouds, and $LDR > 0.15$ for ice clouds. Table 2a shows the quantitative comparison of the cloud
phase identifications between these two classification methods. The numbers represent the counts
of each matched 5-min sample, where the diagonal numbers indicate that both methods are
identifying the same type of cloud phase. In general, the two methods have 89 % agreement on the
phase identification. Secondly, we performed the phase classification directly from microphysical
probes onboard G-1 aircraft during SOCRATES and treated them as 'ground-truth' (Mohrmann et
al., 2021). By changing the time step of sampling to mimic what the radar may observe the cloud
for each range volume, Table 2b shows the statistics of the possibility of the cloud phase that may
be detected by cloud radar. As sampling time increases from 1 second to 30 seconds, more mixed-
phase clouds and fewer single-phase clouds can be observed.

Figure 6 shows the occurrence frequencies of the matched results for mixed-phase clouds (left

panel) and ice clouds (right column). For the classified ice clouds, most of the Doppler spectrum
widths range from 0.08 to 0.16 m s$^{-1}$ and the *LDR* ratios can be up to ~0.5, representing a narrow
range of ice particle size distribution with higher *LDR* ratios. Cloud liquid is identified by low *LDR*
of ~0.11 (Fig. 6d) and high lidar backscatter from 10$^{-5}$ to 10$^{-4}$ m$^{-1}$ sr$^{-1}$ (Fig. 6e). For the classified
mixed-phase clouds, most of the Doppler spectrum width range from 0.15 to 0.35 m s$^{-1}$ and most
of the *LDR* signals are less than 0.4, representing a broad particle size distribution resulting from
the mixture of liquid droplets and ice particles but lower *LDR* ratios. Based on the Doppler velocity,





the mode values for both mixed-phase and ice clouds occur at ~ 0.5 m s$^{-1}$, where the ice particles
are dominant in both types of clouds. The broader particle size distribution with lower *LDR* ratios
for mixed-phase clouds and narrower particle size distribution with higher *LDR* ratios for ice
clouds further corroborate that the classified results from this study are consistent with the
traditional micropulse lidar *LDR* method.
It is important to note that the micropulse lidar signals are usually attenuated and cannot
provide a meaningful signal when the liquid cloud layer is thicker than a couple of hundred meters
(Sassen, 1991). Arctic mixed-phase clouds are typical, with the liquid-dominant layer on the top
of the mixed-phase clouds and the ice-dominant layer underneath. The ceilometer-derived cloud-
base height represents the base of the liquid-dominant layer near the cloud top, while MPL-derived
cloud-base height represents the base of the lower ice-dominant layer (Qiu et al., 2015; Shupe,
2007; Shupe et al., 2005). Over the Arctic, the micropulse lidar signals can penetrate through the
ice-dominant layer to the liquid-dominant layer. However, the mixed-phase clouds over the
Southern Ocean are totally different from those over the Arctic region: they are well mixed (liquid
droplets and ice particles) from cloud base to cloud top, which is found in this study. Thus, the
micropulse lidar signals can be attenuated in the mixed-phase clouds over the Southern Ocean.
Statistical results show that 43 % of micropulse data were attenuated during MARCUS compared
to our classified results.
This classification method is further supported by the onboard cloud radar measurements
during the Southern Ocean Clouds Radiation Aerosol Transport Experimental Study (SOCRATES,
not shown). In that campaign, the reflectivity measurements were usually greater, and the spectrum
widths were much wider when the aircraft observed large ice particles compared to the time
periods when liquid cloud droplets were observed. It is also worth mentioning that about 5.5 % of





single-layered low-level cloud phases cannot be determined when the radar measurements were
not available during MARCUS. Therefore, using our classification method, a total of 6,934 5-min
single-layered low-level cloud samples were determined in this study, including 697 liquid cloud
samples, 3,777 mixed, 1,205 ice, and 1,255 'OTHER' clouds. The category of 'OTHER' clouds
represents more than one phase in each column.
Figure 7 (upper panel) shows the drizzling status for each categorized cloud type, e.g., no rain
(yellow-green), virga (brown) and rain (navy blue). The definition of drizzling status follows the
method in Wu et al., (2015, 2017), which used both ceilometer and cloud radar measurements to
determine the status of MBL clouds under non-drizzling, virga and rain conditions. The
percentages shown below the x-axis represent the portion of drizzling status in each type of clouds,
such as liquid, mixed-phase, ice and 'OTHER' clouds. Figure 7 (bottom panel) also shows the
percentages and vertical distributions of classified liquid, mixed-phase, ice, and 'OTHER' clouds
for each column in the single-layered low-level clouds, represented by different colors. After
classification, the samples in each category are sorted by their $H_{top}$. In detail, Fig. 7 demonstrates
that the mixed-phase clouds dominate the single-layered low-level cloud category with an
occurrence frequency of 54.5 %. The 'OTHER' and ice clouds have similar occurrence frequencies
of 18.1 % and 17.4 %, respectively, while the liquid clouds have the least occurrence frequency of
10.1 %. The liquid topped mixed-phase clouds (included in 'OTHER'), which frequently occur in
the Arctic region (Qiu et al., 2015), are rarely found over the SO. The existence of ice particles in
mixed-phase clouds should strongly depend on the distribution of ice nuclei (IN), whereas spatially
unevenly distributed IN may result in the OTHER type of clouds.
Based on the results in Fig. 7, we draw the following conclusions. Most of the ice clouds are
non-drizzling clouds, and the percentages with virga and drizzle below the cloud base are 12 %





and 15 %, respectively. The percentages of non-drizzling, virga and drizzling mixed-phase clouds

are 50 %, 21 %, and 29 %. The liquid and 'OTHER' clouds have similar percentages, they are 36 %,

25 % and 39 % for liquid clouds, and 35 %, 22 % and 44 % for 'OTHER' clouds. For liquid and

'OTHER' clouds, the drizzling frequencies are independent of $H_{top}$. In contrast, for mixed-phase

and ice clouds, the drizzling frequencies strongly depend on $H_{top}$, i.e., higher drizzling frequencies

occur mostly at higher $H_{top}$.

The properties of single-layered low-level clouds are summarized in Table 3. The liquid clouds

have the lowest $H_{base}$ and $H_{top}$ but more available water vapor than other types of clouds. Since the

'OTHER' clouds are a transitional stage among mixed-phase, liquid and ice clouds, they have the

highest $H_{top}$, deepest cloud layer and largest $LWP$. The ice clouds occur in relatively dry

environments and have the highest $H_{base}$ at 1.218 km. The mixed-phase clouds have similar $H_{base}$,

but lower $H_{top}$, $LWP$ and $PWV$ compared to those of 'OTHER' clouds. Since $LWP$s in mixed-phase

clouds have larger standard deviations, which implies that SLW is more common at higher $LWP$s

and ice is more common at lower $LWP$s.

**4.2. Meridional variations of cloud properties**

Figure 8 shows the meridional variation in single-layered low-level cloud properties during

MARCUS. As illustrated in Fig. 8a, the meridional distributions of $H_{base}$, $H_{top}$ and $\Delta H$ are nearly

independent of latitude, however, their corresponding temperatures ($T_{base}$ and $T_{top}$) increased about

8 K from 69 °S to 43 °S, though there were slight fluctuations. These results suggest that the cloud

and sea surface temperatures have minimal impact on the cloud boundaries over the SO, which is

consistent with the findings in McFarquhar et al. (2016). The meridional variation of $LWP$s mimics

those of $T_{base}$ and $T_{top}$, with an increasing trend from south to north. It is important to point out that

a big drop in $LWP$ at ~50 °S results from fewer occurrences of low-level clouds there, indicating



that the cloud samples at some latitudes are not statistically significant. The atmospheric *PWV*
increased dramatically from ~ 5 mm at 69 °S to ~18 mm at 43 °S, presumably due to increased sea
surface and atmospheric temperatures.
Figure 9 shows the latitudinal and meridional distributions of categorized liquid, mixed-phase,
ice and 'OTHER' in single-layered low-level clouds over the SO during MARCUS. Each circle
represents the exact location and time along the ship track. Mixed-phase clouds occurred
everywhere over the SO during the MARCUS field campaign and became dominant in November,
December and February. Liquid clouds dominated in March, while ice clouds dominated in
January. The 'OTHER' clouds are a kind of transitional phase falling in between the mixed-phase
and ice/liquid clouds because there are no stand-alone occurrences in any month during MARCUS.
**4.3 Vertical distribution of cloud properties**
The vertical distributions of classified liquid, mixed-phase, and ice clouds are presented in Figs.
10-12. The focus of this section will be comparisons of cloud properties between the north (NSO)
and south (SSO) regions of the domain. Figure 10a shows the vertical distributions of liquid clouds,
which were capped at ~ 1.6 km, mostly in the marine boundary layer. The vertical occurrence
frequencies are up to 27 % over NSO, while they were less than 4 % over SSO, i.e., liquid clouds
occurred fairly often over the mid-latitude region, but very few occurred over the Polar region. On
the contrary, the occurrence frequencies of mixed-phase clouds between NSO and SSO are
opposite to liquid clouds, as illustrated in Fig. 10b, though the differences are not so obvious.
Mixed-phase clouds increased with altitude until ~1.6 km, then decreased monotonically towards
3 km. The highest frequencies were ~37 % at 0.6 km over SSO and ~27 % at 1.5 km over NSO.
The vertical distributions of ice clouds are similar to those of mixed-phase clouds (Fig. 10c).
However, there were no significant differences between NSO and SSO. It is worth mentioning that



the vertical distributions of mixed-phased clouds over SO are quite different to those from DOE
ARM Northern Slope Alaska (NSA) site where the liquid topped mixed-phase low-level clouds
are common (e.g., Qiu et al., 2015).
To further investigate the vertical distributions of classified liquid, mixed-phase, and ice clouds
over NSO and SSO, we plot the normalized vertical distributions (cloud base as 0, cloud top as 1)
of radar reflectivity, Doppler velocity and spectrum width in Figs. 11 and 12, respectively. In this
study, the threshold of -50 dBZ was used to determine the cloud boundary over the SO instead of
the threshold of -40 dBZ radar reflectivity used at the ARM ENA site (Dong et al., 2014). If we
used the threshold of -40 dBZ over the SO, then there would be only 73 % cloud samples available
for this study. About 9.6 % of radar reflectivities during MARCUS are less than -50 dBz for all
single-layered low-level cloud samples. Thus all radar parameters used in this study are based on
90.4 % of the radar measurements with reflectivity greater than -50 dBz.
Figures 11a-11c represent the normalized vertical distributions of radar reflectivity, Doppler
velocity and spectrum width of liquid clouds. Liquid clouds had the lowest reflectivity near the
cloud top because of cloud-top entrainment., The reflectivity had a nearly constant median value
of ~ -22 dBZ from near cloud top height (~ 0.8 for normalized height) of the cloud layer to the
cloud base. Most of the reflectivities were less than -15 dBz, which is the threshold to distinguish
cloud and drizzle-sized particles in each radar range volume (Wu et al., 2020). Most of the Doppler
velocities were greater than 0.0 m s$^{-1}$, indicating that downwelling motion is dominant in liquid
clouds. The profiles of Doppler velocity and spectrum width increased smoothly from the cloud
top to base, suggesting that larger cloud droplets and broader size distributions exist near the cloud
base, which is attributable to more drizzle drops near the cloud base, as illustrated in Fig. 7.





The vertical distributions of mixed-phase clouds in Figs. 11d-11f are similar to those of liquid
clouds. The more occurrences of larger reflectivity measurements and larger median values of
spectrum width near the cloud base are most likely due to the presence of moderate ice particles
and/or drizzle drops. The nearly same median values of reflectivity, Doppler velocity and spectrum
width (but slightly larger standard deviations in each level in mixed-phase clouds) in both liquid
and mixed-phase clouds suggest that the ice particle sizes in mixed-phase clouds are comparable
to cloud droplets and drizzle drops. The nearly uniform vertical distributions of Doppler velocity
and spectrum width indicate well-mixed liquid cloud droplets and ice particles throughout the
cloud layer in the mixed-phase clouds over NSO.
Ice clouds had much lower reflectivity and narrower spectrum width than liquid and mixed-
phase clouds, as shown in Figs. 11g-11i. Almost all reflectivity measurements were less than -25
dBz with a median value of -35 dBz at the cloud base, resulting from small or moderate ice
particles but much lower concentration. A nearly constant Doppler velocity within the cloud layer
further supports the discussion of mixed-phase clouds above, i.e., the ice particle sizes are
independent of cloud height and comparable to liquid cloud droplets in the low-level clouds over
the SO. Because there are no mechanisms for growing large ice particles in such shallow ice clouds,
the accretion process cannot take place. From the statistical results in Fig. 7, these ice particles
have relatively little chance to become virga or raindrops and usually dissipate or transition to
other types of clouds.
Since there are not enough liquid cloud samples over the Polar region, only the mixed-phase
and ice clouds results are plotted in Fig. 12. Compared to the vertical distributions of ice clouds
over NSO, the median values of reflectivity and Doppler spectrum width over SSO were lower
and narrower, indicating a lack of large ice particles in the Polar region. The small ice particles in





the Polar region were also reflected in their mixed-phase clouds. Compared to the vertical
distributions of the mixed-phased clouds over NSO, the median values of reflectivity and Doppler
spectrum width over SSO were dramatically lower (-35 dBz at SSO vs. -22 dBz at NSO; 0.25 m
$s^{-1}$ at SSO vs. 0.32 m $s^{-1}$ at NSO). Figure 12 illustrates that the ice particle sizes over SSO are
smaller, their size distributions are narrower than those over NSO, indicating of lack of large ice
particles over SSO.

**5. Summary and Conclusions**
In this study, we presented the statistical results of clouds over the Southern Ocean (SO) and
the northern (NSO) and southern (SSO) parts during MARCUS IOP. We used the method
developed in Xi et al. (2010) to calculate the occurrence frequencies of different types of clouds
and their corresponding cloud macrophysical properties. We developed a new method to classify
liquid, mixed-phased, and ice clouds in the single-layered low-level clouds as well as their
corresponding drizzling status. Lastly, we explored the meridional and vertical distributions of
these classified cloud properties. Analysis of the MARCUS cloud properties has yielded the
following conclusions.
1.     The total cloud fractions (*CF*s) were 77.9 %, 67.6 %, and 90.3 % for the entire
domain, NSO and SSO, respectively, indicating that 22.7 % more clouds occurred in the Polar
region than in the mid-latitude region. The SSO had more clouds under 7 km, while the NSO
had more clouds above 7 km. Below 3 km, the occurrence frequencies of clouds over NSO
decrease dramatically from 37 % at an altitude of ~700 m to 16 % at 3 km, which is similar to
the vertical distributions of the low-level clouds over some Northern Hemisphere mid-latitude
regions, such as Eastern North Atlantic.
2.    The single-layered low-level (LOW), deep cumulus or multi-layered (HML), and
shallow cumulus (MOL) clouds are the three dominant types of clouds over the SO. Comparing
the clouds between NSO and SSO, all types of clouds in SSO are higher than those in NSO
except HOL. The LOW fraction, thickness, LWP over both NSO and SSO are comparable to
each other. The mean $LWP$s for low clouds over NSO, such as LOW, MOL and HOL, range
from ~130 to 150 g m$^{-2}$, while the mean $LWP$s for shallow and deep convective clouds, such
as MOL and HML, are two times (~270 g m$^{-2}$) higher than the same types of clouds. The mean
$LWP$s of clouds over SSO are much lower than the $LWP$s over NSO. Over the Southern Ocean,
the single-layered or contiguous clouds usually have higher liquid water paths than their
counterpart of multi-layered or non-contiguous clouds. There are more non-contiguous HML
and HOM than contiguous ones.
3.    A new method was developed to classify liquid, mixed-phase and ice clouds in the
single-layered low-level clouds (LOW) based on comprehensive ground-based observations.
The mixed-phase clouds are dominant in the single-layered low-level cloud category with an
occurrence frequency of 54.5 %. The 'OTHER' and ice clouds had similar occurrence
frequencies of 18.1 % and 17.4 %, respectively, while the liquid clouds had the least occurrence
frequency of 10.1 %. The percentages of non-drizzling, virga and drizzling for mixed-phase
clouds were 50 %, 21 %, and 29 %, and the drizzling frequencies of mixed-phase clouds
strongly depend on $H_{top}$, that is, higher drizzling frequencies occurred mostly at higher $H_{top}$.
4.    The meridional distributions of $H_{base}$, $H_{top}$ and $\Delta H$ are nearly independent on
latitude. However, their corresponding temperatures increased about 8 K from 69 °S to 43 °S.
The meridional variation of $LWP$s mimics that of cloud temperatures, having an increasing
trend from south to north. The mean $PWV$ increased dramatically from ~ 5 mm at 69 °S to ~18

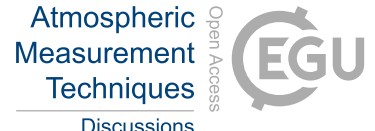
mm at 43 °S due to increased sea surface and atmospheric temperatures. More liquid clouds
occurred over NSO but very few occurred over SSO, whereas more mixed-phase clouds
occurred over SSO than over NSO. There were no significant differences in ice clouds
occurrences between NSO and SSO.
5.    The nearly same median values of reflectivity, Doppler velocity and spectrum
width in both liquid and mixed-phase clouds over NSO suggest that the ice particle sizes in
mixed-phase clouds are comparable to cloud droplets and drizzle drops. The uniform vertical
distributions of Doppler velocity and spectrum width suggest well-mixed liquid cloud droplets
and ice particles throughout the cloud layer in the mixed-phase clouds over NSO, which are
quite different from those over the DOE ARM NSA site where the liquid topped mixed-phase
low-level clouds are common. The median values of reflectivity and Doppler spectrum width
over SSO were lower and narrower than those over NSO, indicating lack of large ice particles
in the polar region.
These results provide comprehensive statistical properties of all clouds over the SO during
MARCUS, including the occurrence frequencies of different types of clouds and their
corresponding cloud macrophysical properties. We also examined the meridional and vertical
distributions of the classified cloud properties. These statistics can be used as ground truth to
evaluate satellite retrieved cloud properties and model simulations over the SO. The results of this
study will help to advance our understanding of these clouds, which may lead to improved model
simulations over the SO as well as a better representation of global climate.

*Data availability.* Data used in this study can be accessed from the DOE ARM's Data Discovery at
https://adc.arm.gov/discovery/



*Author contributions.* The idea of this study is discussed by BX, XD, and XZ. BX and XZ performed
the analyses and BX wrote the manuscript. BX, XD, XZ and PW participated in scientific discussions
and provided substantial comments and edits on the paper.

*Competing interests.* The authors declare that they have no conflict of interest.

*Acknowledgements.* The ground-based measurements were obtained from the Atmospheric
Radiation Measurement (ARM) Program sponsored by the U.S. Department of Energy (DOE)
Office of Energy Research, Office of Health and Environmental Research, and Environmental
Sciences Division. The data can be downloaded from http://www.archive.arm.gov/. Researchers
were supported by the NSF project under grant AGS-2031750 at the University of Arizona.
Specially thanks to Mr. Xingyu Zhang for providing analysis from CDP and 2DS microphysical
sensors during SOCRATES and Dr. Dale Ward at the University of Arizona for proofreading this
manuscript.

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





**Table 1. Minimum cloud base heights ($H_{base}$), maximum cloud top heights ($H_{top}$), and liquid water paths ($LWP$s) (all samples, single-layered, multilayered) of all seven types of clouds over the Southern Ocean. Cloud heights have unit of kilometer, and $LWP$ has unit of g m$^{-2}$.**

|  | LOW | MID | MOL | HGH | HOM | HML | HOL |
|---|---|---|---|---|---|---|---|
| **$H_{base}$ ± std** | 0.92 ± 0.57 | 4.14 ± 0.61 | 1.37 ± 0.96 | 8.51 ± 2.23 | 4.70 ± 0.80 | 1.22 ± 0.98 | 1.14 ± 1.12 |
| min, max | 0.06, 2.86 | 3.00, 5.84 | 0.06, 5.27 | 6.00, 18.67 | 3.01, 7.72 | 0.06, 7.81 | 0.07, 10.37 |
| **$H_{top}$ ± std** | 1.62 ± 0.63 | 4.88 ± 0.68 | 4.29 ± 0.89 | 9.75 ± 2.13 | 7.93 ± 1.27 | 7.81 ± 1.35 | 8.93 ± 1.66 |
| min, max | 0.29, 3.0 | 3.17, 6.0 | 1.39, 5.99 | 6.20,18.79 | 5.47, 17.98 | 3.62,17.38 | 1.79, 17.56 |
| **$LWP$ ± std** | 122.4±134.2 | 86.7±124.5 | 168.7±236.7 | / | 40.9±40.8 | 169.2±238.4 | 129.8±202. |
| Max $LWP$ | 1470.8 | 501.1 | 1937.1 | / | 345.7 | 1819.3 | 1785.2 |
| **$LWP$ ± std (single layer)** | 126.6±138.1 | 88.7±128.9 | 193.1±271.9 | / | 48.7±51.7 | 270.8±349.5 | / |
| max | 1470.8 | 501.1 | 1937.1 | / | 345.7 | 1819.3 | / |
| **LWP ± std (multi-layer)** | 96.2±103.4 | 77.2±109.2 | 139.0±180.7 | / | 32.3±21.3 | 148.4±202.4 | 129.8±202. |
| max | 842.3 | 305.6 | 1830.2 | / | 86.8 | 1690.7 | 1785.2 |
| **Multi-layer percentage %** | 18.1 | 39.6 | 50.0 | 44.9 | 73.1 | 77.7 | 100 |

\* The definition of the cloud types as follow: LOW ($H_{base}$ and $H_{top} \leq 3\ km$); MID ($H_{base} > 3\ km$ and $H_{top} \leq 6\ km$); HGH ($H_{base} > 6\ km$); MOL ($H_{base} < 3\ km$ and $H_{top} \leq 6\ km$); HOM ($3\ km < H_{base} < 6\ km$ and $H_{top} > 6\ km$); HML ($H_{base} < 3\ km$, $H_{top} \geq 6\ km$ with a MID layer); HOL (LOW and HGH appear at the same time).



**Table 2a. Comparison of cloud phase identifications between our classification method and Shupe et al. (2005) method in each 5-min measurements, the unit is number of 5-min samples.**

| Shupe / this study | Liquid (this study) | Mixed-phase (this study | Ice (this study) |
|---|---|---|---|
| Liquid | 468 | 490 | 0 |
| Mixed-phase | 98 | 3840 | 0 |
| Ice | 81 | 0 | 1195 |

*Numbers denote the cloud sample classifications between two methods. For example, the number 98 denote a total of 98 samples are classified as Mixed-phase using Shupe's method, while are classified as Liquid using this study's method.

**Table 2b. The cloud phase partitioning from CDP and 2DS during SOCRATES. Cloud Droplet Probe (CDP) measures particle size from 2 to 50 um in diameter; Two-Dimensional Stereo Probe (2DS) measures particle size from 50 to 5000 um in diameter.**

| Phase partitioning | 1 second | 10 seconds | 30 seconds |
|---|---|---|---|
| Samples # | 27280 | 2255 | 836 |
| Liquid, % | 58.8 | 26.2 | 18.8 |
| Mixed-phase, % | 38.9 | 69.1 | 77.0 |
| Ice, % | 2.3 | 4.7 | 4.2 |



**Table 3. Liquid, mixed, ice and OTHER phases of cloud properties within the single-layered low-level clouds**

| Phase | Samples | $H_{base}$, km | $H_{top}$, km | $\Delta H$, km | $LWP$, g m$^{-2}$ | $PWV$, mm |
|---|---|---|---|---|---|---|
| **Liquid** | 697 | 0.424±0.204 | 1.327±0.242 | 0.903 | 113.6±90.1 | 15.7±3.5 |
| **Mixed** | 3777 | 0.834±0.465 | 1.434±0.617 | 0.587 | 119.7±136.6 | 8.9±5.0 |
| **Ice** | 1205 | 1.218±0.635 | 1.737±0.651 | 0.519 | 0 | 8.4±4.5 |
| **OTHER** | 1255 | 0.700±0.454 | 1.774±0.571 | 1.074 | 141.9±137.5 | 11.4±5.9 |



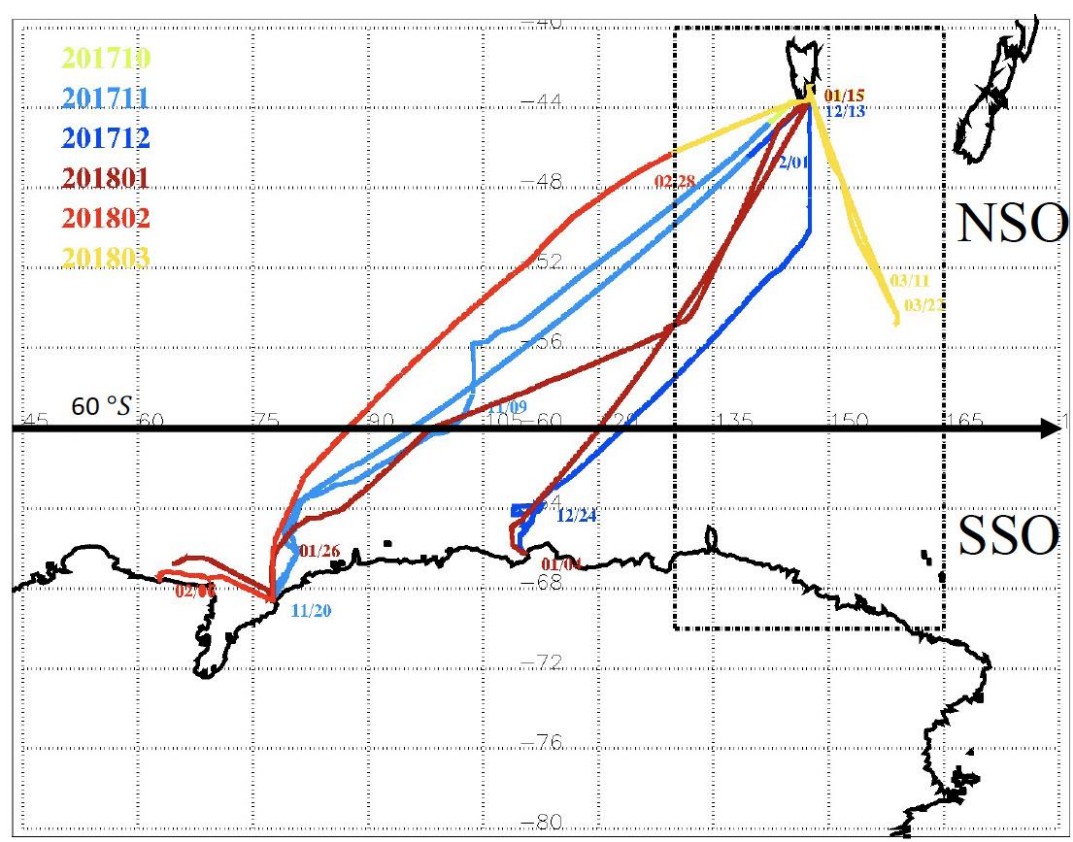

**Figure 1**. Shiptrack measurements between Hobart, Australia and Antarctica. Different colors represent different month's shiptracks from Oct. 29, 2017 to Mar. 23, 2018 during MARCUS. Along the shiptracks, the study domain is separated into northern (NSO) and southern (SSO) parts of the Southern Ocean with a demarcation line of 60 °S in order to study the clouds over the mid-latitudes (North of 60 °S) and Polar region (South of 60 °S). The black dotted rectangle represents the SOCRATES study domain. Some of the dates have labeled along the shiptracks, which can indicate the direction of the ship traveled.



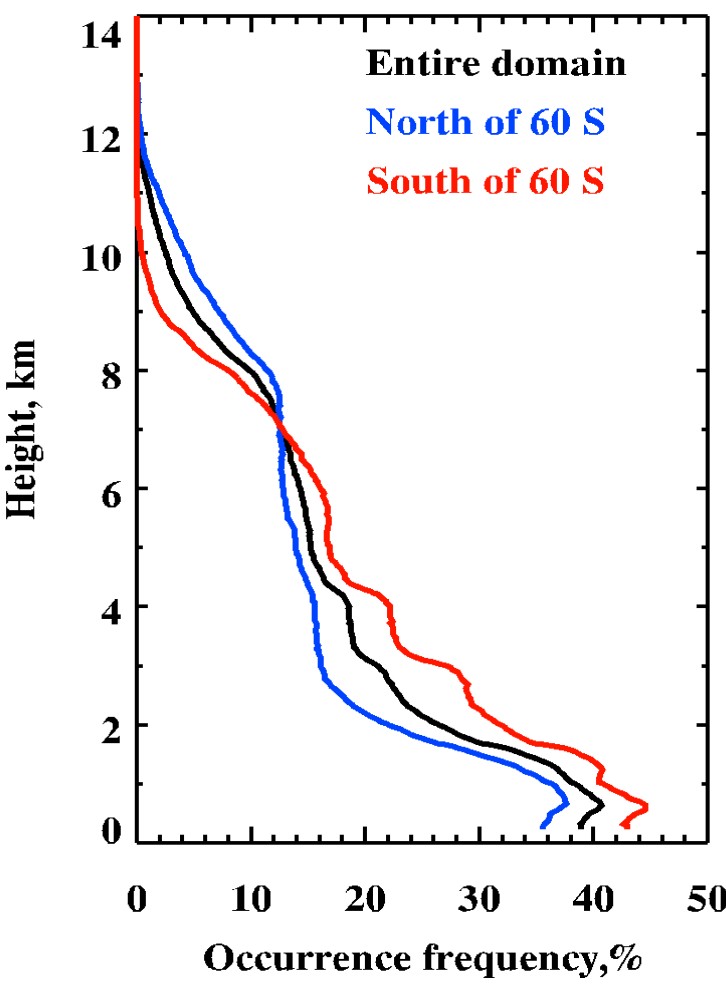

**Figure 2**. Mean vertical distributions of total clouds derived from ARM radar-lidar observations with a 5-min temporal resolution and a 30-m vertical resolution during MARCUS.

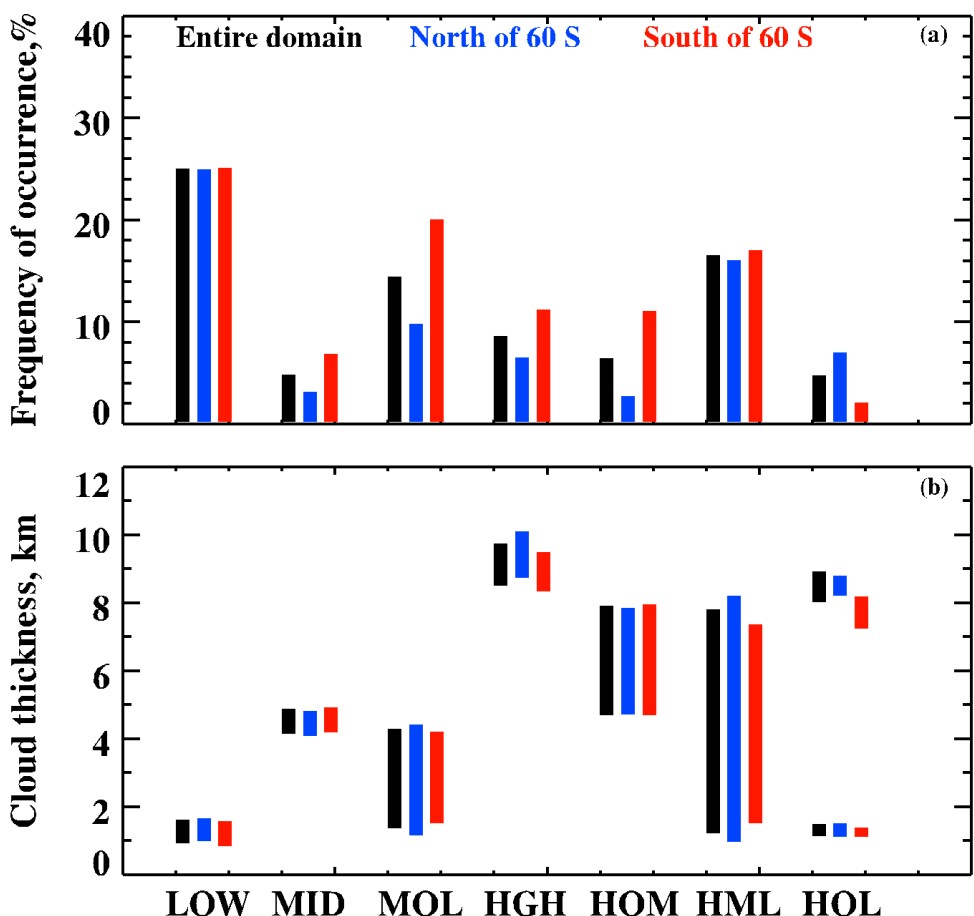

**Figure 3**. (a) Occurrence frequencies of categorized clouds by their vertical structures. LOW, single‑layered low clouds ($H_{base}$ and $H_{top} \leq 3\ km$); MID, single-layered middle clouds ($H_{base} > 3\ km$ and $H_{top} \leq 6\ km$); MOL, MID over LOW ($H_{base} < 3\ km$ and $H_{top} \leq 6\ km$); HGH, single-layered high clouds ($H_{base} > 6\ km$); HOM, HGH over MID ($3\ km < H_{base} < 6\ km$ and $H_{top} > 6\ km$); HML, HGH over MID and LOW ($H_{base} < 3\ km$, $H_{top} \geq 6\ km$ with a MID layer); and HOL, HGH over LOW (LOW and HGH appear at the same time). (b) Cloud thickness for each type of clouds (bar), the top and bottom of the bar represent the maximum cloud-top and minimum cloud-base heights, respectively. Black, blue, and red bars represent the entire domain (Lat:41-69 °S; Long: 60-160° E), north of 60 °S (NSO), and south of 60°S (SSO), respectively, during the MARCUS field campaign (10/2017-3/2018).



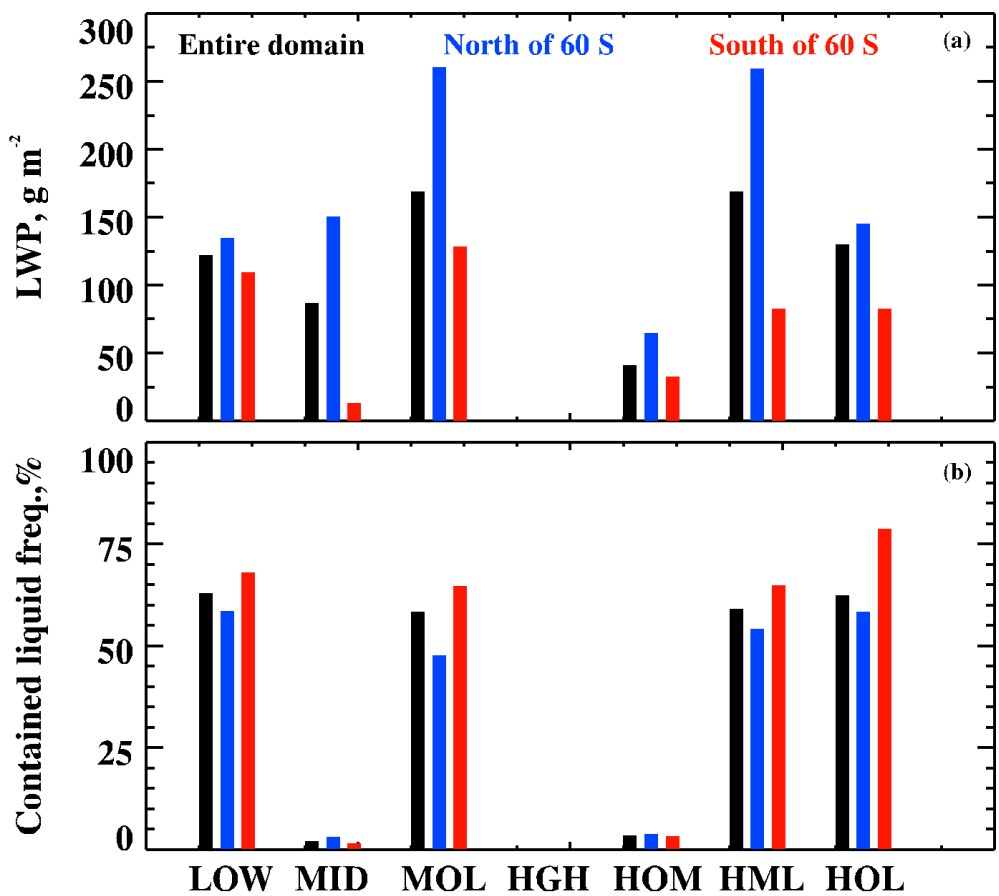

**Figure 4**. (a) Cloud liquid water paths (*LWP*s) retrieved from microwave radiometer (MWR) measured brightness temperature using a physical retrieval method for each type of cloud. (b) The occurrence frequencies of LWPs> 10 gm$^{-2}$ for each type of clouds


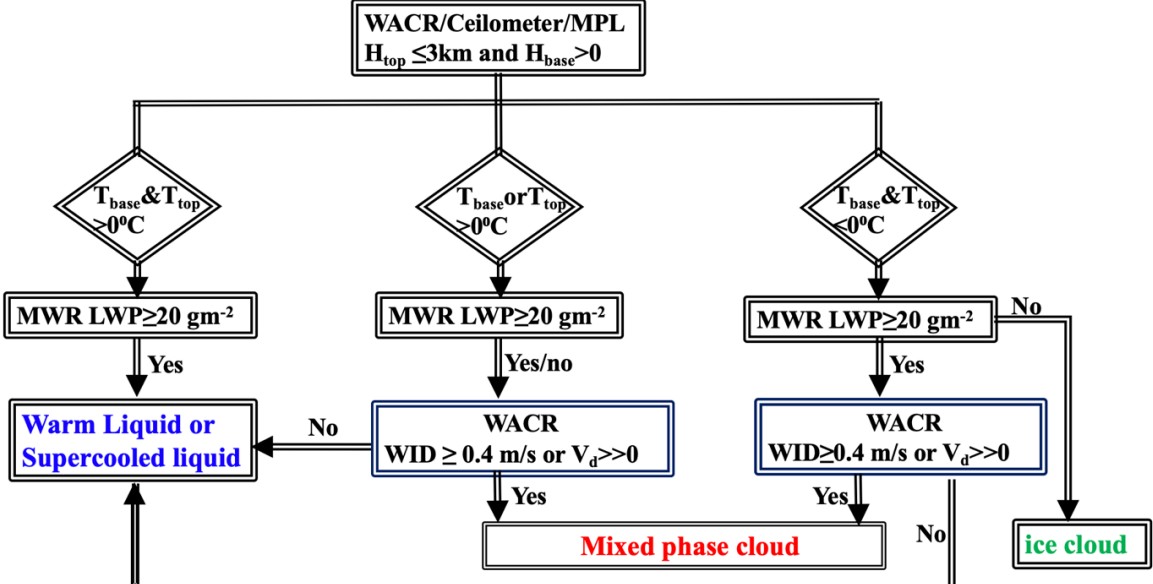

**Figure 5.** A flow chart for phase classification of single-layered low-level clouds. W-Band (95 GHz) ARM Cloud Radar (WACR) provides radar spectrum width (*WID*) and Doppler velocity ($V_d$).

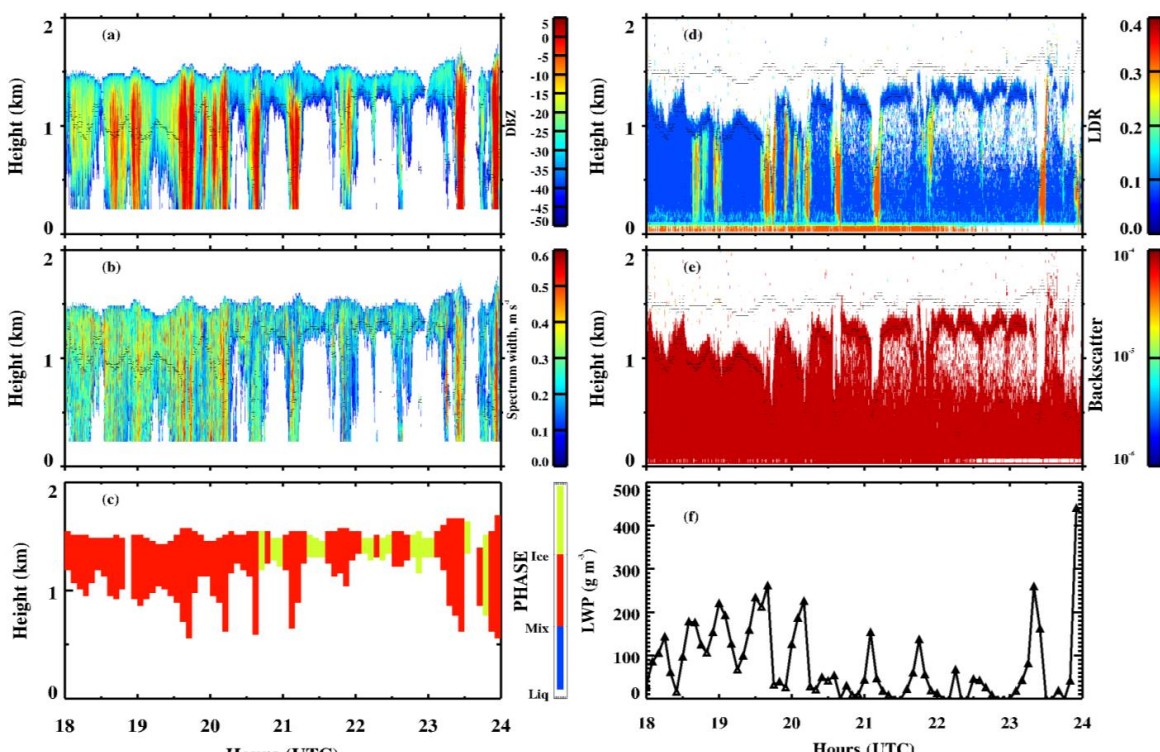

**Figure 6**. A case study that shows our phase classification (left column) and Micropulse Lidar linear depolarization ratios (*LDR*) and backscatter. W-Band (95 GHz) ARM Cloud Radar (WACR) reflectivity shows in (a) and spectrum width shows in (b); The phase classification shows in (c); MPL *LDR* shows in (d) and backscatter shows in (e); and *LWP* shows in (f).

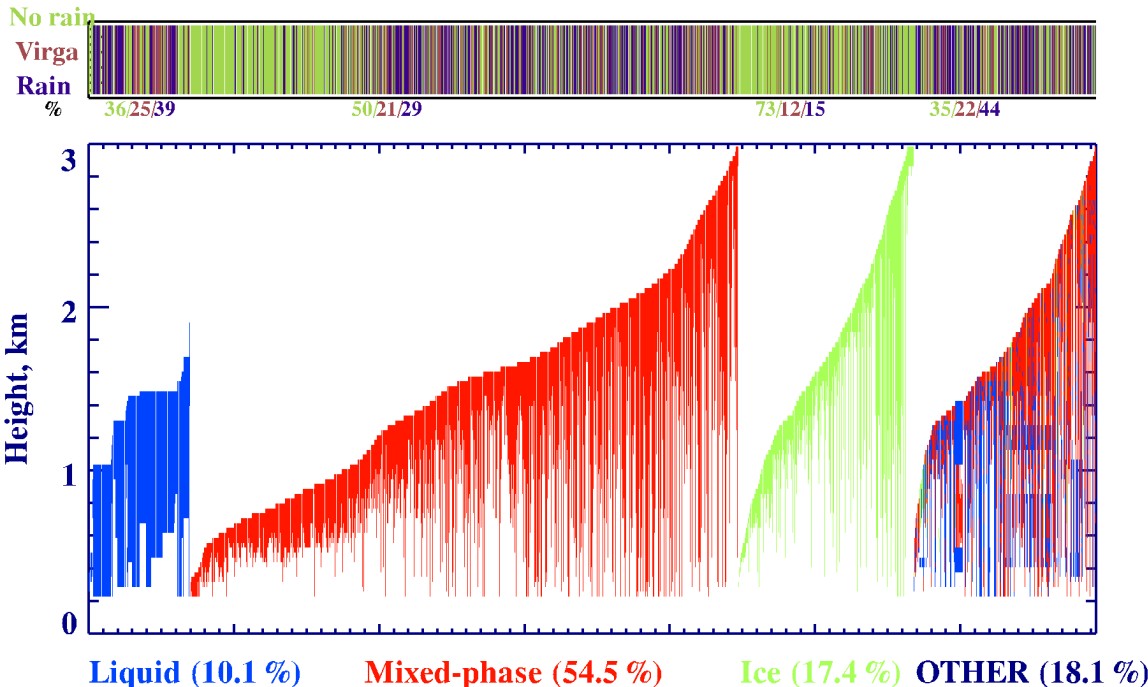

**Figure 7**. (Upper Panel) The drizzling status for each categorized cloud type, e.g., no rain (yellow-green), virga (brown) and rain (navy blue), the percentages shown below the x-axis represent the portion of drizzling in each type of clouds; And (Bottom Panel) the percentages and vertical distributions of classified liquid, mixed-phase, ice, and 'OTHER' clouds for each column in the single-layered low-level clouds, represented by different colors, over the entire domain during MARCUS. Each line represents one 5-min sample.

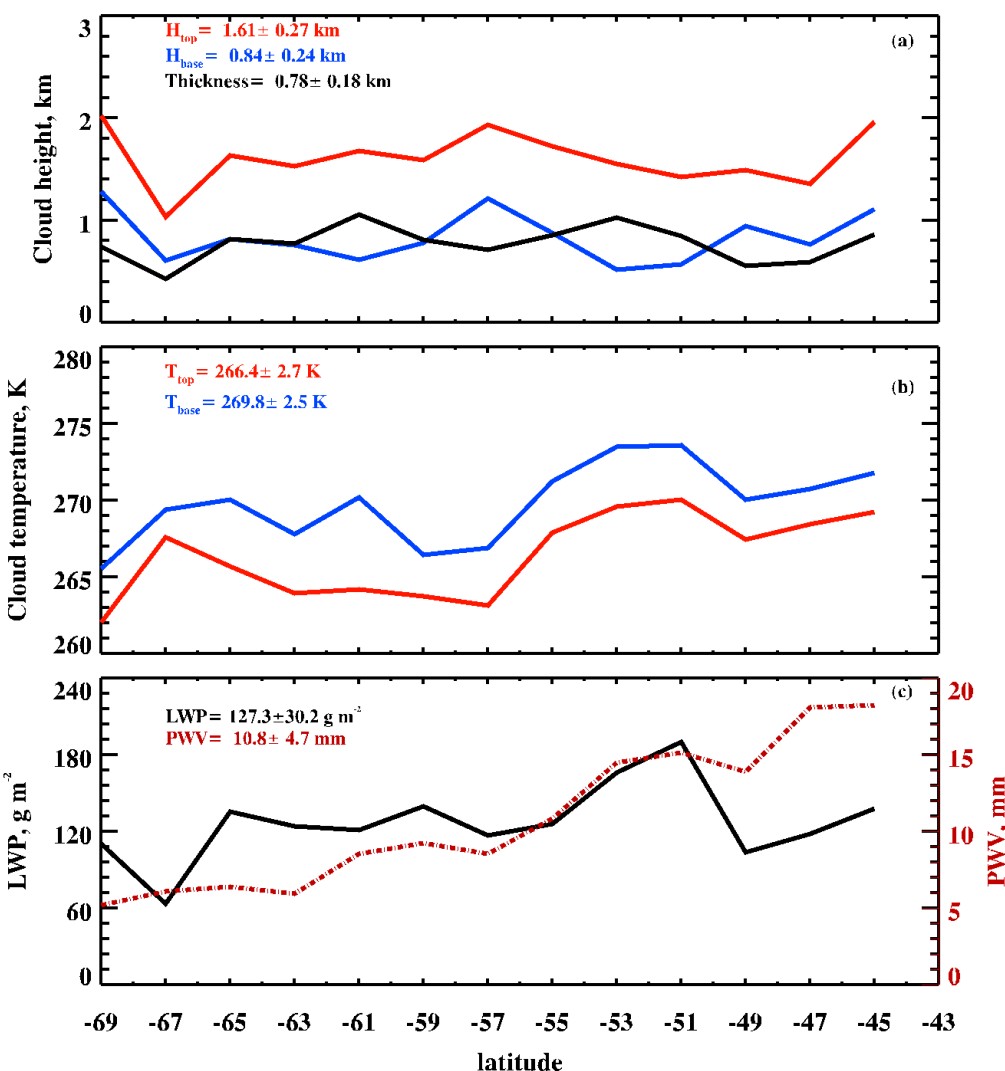

**Figure 8**. Meridional variations of single-layered low-level cloud properties: (a) cloud-base ($H_{base}$) and -top ($H_{top}$) heights, and cloud thickness ($\Delta H$), (b) cloud-base ($T_{base}$) and -top ($T_{top}$) temperatures, and (c) cloud liquid water path (*LWP*) and precipitable water vapor (PWV) over the entire domain during MARCUS.

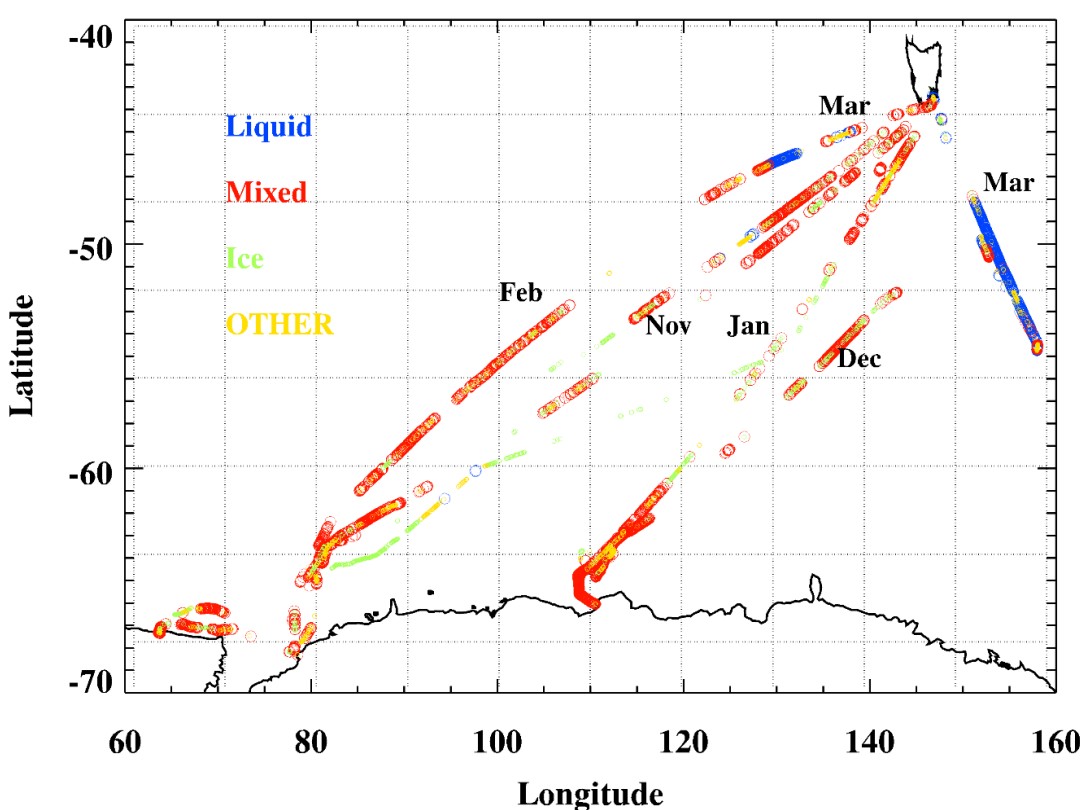

**Figure 9**. The latitudinal and longitudinal distributions of classified mixed-phase, liquid, and ice clouds in the single-layered low-level clouds. The liquid (blue), mixed (red), ice (light green), and OTHER (yellow) are shown along each shiptrack from October 2017 to March 2018 during MARCUS.

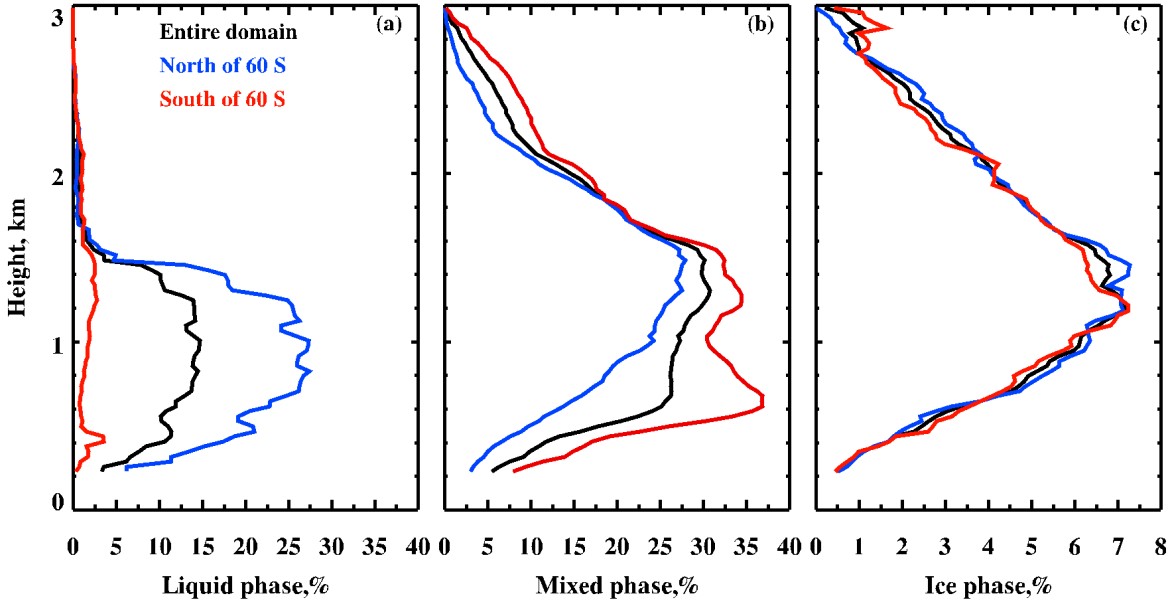

**Figure 10**. Occurrence frequencies of classified mixed-phase, liquid, and ice clouds over the entire domain (black), North of 60 °S (blue) and South of 60 °S during MARCUS.



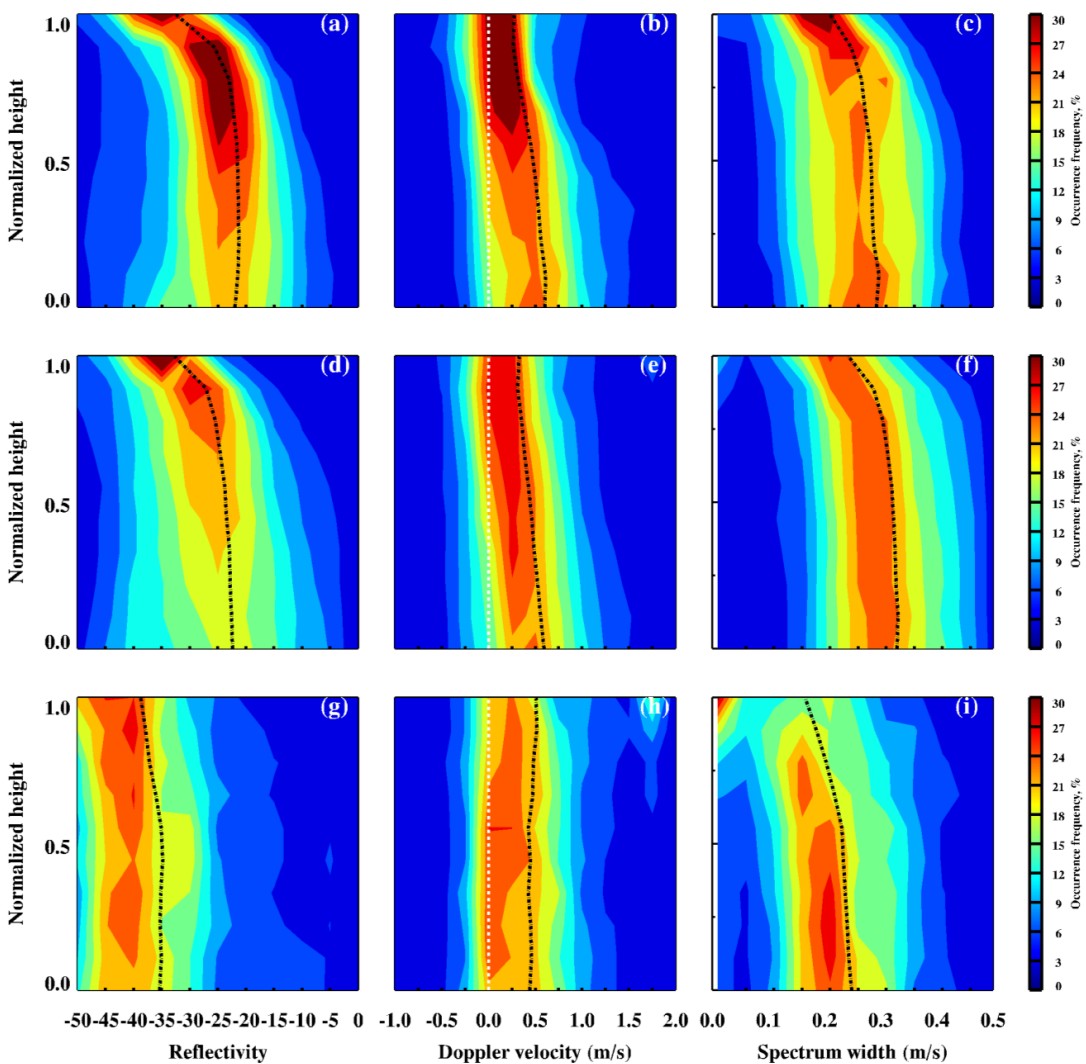

**Figure 11**. Normalized vertical distributions of a) radar reflectivity, b) Doppler velocity and c) spectrum width for the classified liquid (upper panel), mixed-phase (d to f, middle panel) and ice (g to i, bottom panel) clouds over the North of 60 °S during MARCUS IOP. Normalized height is defined as $= \frac{H - H_{base}}{H_{top} - H_{base}}$ where cloud base is denoted as 0 and cloud top is 1. The black lines represent the median values and the white lines in Doppler velocity represent the reference of 0.0 m s$^{-1}$.



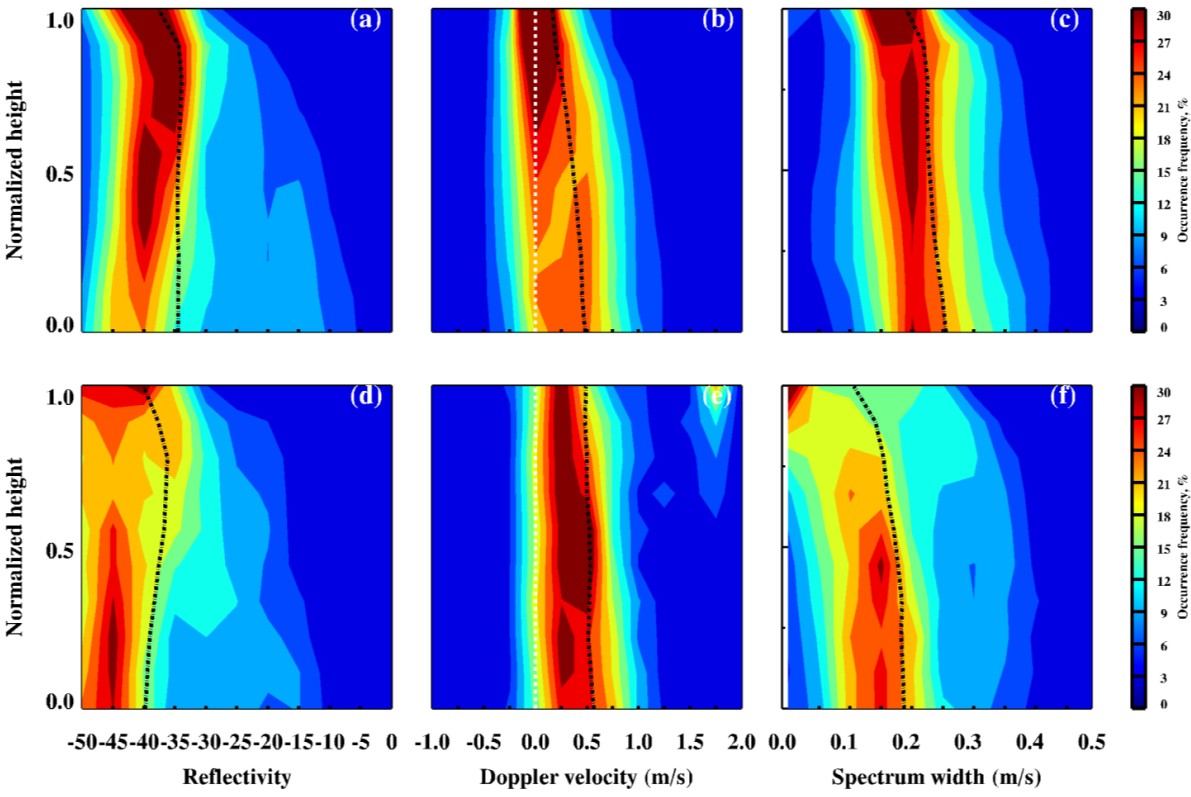

**Figure 12**. Same as Fig. 11 but only for mixed-phase (a to c, upper panel) and ice (d to f, bottom panel) over the south of 60 °S during MARCUS.