# Peer review of "Cloud phase and macrophysical properties over the Southern Ocean during the MARCUS field campaign"

_Atmospheric Measurement Techniques, 2021_

## Referee Comment (RC1)

**Comments:**

Line 1: The title specifies this study as "Cloud properties over the Southern Ocean During the MARCUS field campaign". The "cloud properties" is a broad description including macrophysics, microphysics and dynamics, etc. However, the focus of this work is on the investigation of the macrophysics distribution and the cloud phase characteristics. I suggest using a more specific title to be consistent with the scope of the manuscript.

Line 17: It seems that cloud type is classified with ambiguity. Low-level clouds are identified based on the cloud altitude, while shallow cumulus clouds are classified with their morphology and height. I would prefer to consider the shallow cumulus are a subgroup of low-level clouds instead of a separated group. In the manuscript the authors only use cloud height to identify the shallow/deep cumulus, which is not sufficient. I suggest a thorough revision of these terminologies and use them consistent in the manuscript to avoid any confusion. Specifically, I suggest using low/middle/high clouds and avoid the "cumulus" category unless a more specific cumulus classification algorithm is being added in the manuscript.

Line 134: Does the cloud occurrence frequency shown in Figure 2 sensitive to the length of the time window? If so, why use 5 min? Line 188 also mentions the relative low cloud fractions compared to Satellite results is due to the inconsistent resolution. Adapting the wind speed from sounding, a reasonable "equivalent temporal resolution" of the surface-based radar may be obtained. For example, considering the windspeed as 10m/s, a 2-grid box(~200km) corresponding to approximately 5.5 hours of the radar observation (not considering the speed of ship). Thus, does the cloud frequency estimated from a longer time window is closer to the spaceborne results? I'm not suggesting performing a sensitive test for Fig.2, but just want to remind the authors that it is necessary to discuss this issue furthermore for the presented results to be better used for model validation, as the authors mentioned in the introduction.

Line 200: How close? What is the specific CF from Listowski et al, 2019?

Line 214: I also mentioned in the previous comment. The classified cloud type is solely based on cloud heigh, it is not clear why the MOL is related to shallow cumulus.

Line 215: Are you refereeing to occurrence frequency is higher?

Line 226: For Figure 3. Why use the deepest cloud thickness instead of the mean/median thickness? It is hard to interpret the thickness distribution in a statistical perspective as the deepest thickness may be the results of two "outliers" of the cloud in each group. Besides, I suggest using the box plot to present the results shown in Fig. 3b.

Line 232: How the stability is estimated and what is the value for NSO?

Line 235: …HGH and HOM peak "are" less than 10 g m$^{-2}$

Line 241: Same as before, the shallow convective clouds are not clearly identified.

Line 248: Table1: Compared to the mean LWP, the std (uncertainty) is too large and indicates the result is statistically insignificant. This may be due to the insufficient MWR observations being collected in the analysis. Have you considered the period when MWR is wet and thus the LWP retrieval becomes unreliable? The maximum LWP value (1937 g m$^{-2}$) seems too large to be accurately retrieved by MWR.

Line 250: Does the "non-contiguous clouds" refers to multi-layer clouds?

Line 251-252: Please add the corresponding statistics in Table 1 to support this statement.

Line 266: Same as before, I suggest using the mean cloud thickness instead of the "maximum" cloud thickness before making further statements.

Line 288:290: What is the meaning of the "majority of WID?" Why not using the mean/median WID? What is the radar temporal resolution?

Line 292: And or "or"?

Line 289L Any supporting information regarding on the chosen of 0.4m/s of WID?

Line 288-Line 294: Please rephrase the description of the classification algorithm, which is not consistent with Figure 5.

Line 293: LWP threshold (10 gm$^{-2}$) from the manuscript and Figure 5 (20 g m$^{-2}$) is not consistent.

Line 305-310: Not quite understand why this analysis is helpful to validate the proposed classification algorithm.

Line 315: I cannot tell the backscatter difference for Fig.6e, please adjust the color bar range to make the figure more distinguishable. Also, the backscatter unit is missing.

Line 321-323: From Fig. 6c, the identified ice period from radar around 22 UTC corresponding to low LDR value in Fig 6.d, which seems contradictory to this statement.

Line 337-339: It is necessary to present the SOCRATES results if this can further support the classification output. One thing should be considered for the onboard radar is that the Doppler spectrum width may be much higher than the surface-based radar as for the enhanced Doppler spectrum broadening caused by the moving of aircraft.

Line 345-346: What is the difference between mixed-phase and "other" clouds?

Line 364: Drizzle is one kind of the liquid water droplets; why drizzle occurs in ice cloud?

Line 411-Line 413: This comparison only make scene if the cloud phase classification algorithm is same for these two studies.

Line 417: Radar reflectivity need to be calibrated before used for analysis. (Mace., 2021)

Line 429: This statement is valid only if the vertical air velocity is negligible compared to the terminal velocity of cloud/ice particles.

---

## Referee Comment (RC2)

General comment:

This study presents a statistical analysis of cloud types, their occurrence and macrophysical properties over a Southern Ocean region (and separately its northern and southern parts), based on measurements from the MARCUS campaign. In most cases, presentation of results, discussion and comparison with other studies is adequate. The results are relevant for AMT and can be useful for evaluation studies of satellite retrievals and model simulations. For these reasons, I recommend that this manuscript is accepted for publication after major revisions. These regard mainly structural issues (particularly in Section 2) and several points throughout the text that need to be clarified.

Specific comments:

Section 1:

Lines 97-98: "which in-situ … this analysis". This part of the sentence is not clear. Do you mean that in-situ measurements from SOCRATES were used here as reference? Please clarify and rephrase.

Lines 99-102: while the objectives of the MARCUS campaign are clearly stated in this sentence, it is not clear what the focus of this study is. While it is described in the next paragraph, this sentence needs rephrasing.

Section 2 general comment: It was difficult for me to follow the text in Section 2. The authors describe data sets, algorithms and methods in an unclear and mixed way. Please provide a more concise description of data sets and methods by considering the following suggestions:

- It would be helpful for the reader if you provide a table with all the cloud parameters measured and analyzed in this study.

- For each cloud parameter, please provide a short description of the instrument and method used for its measurement or retrieval.

Line 118: please provide a reference or short description for the AMF2 instruments suite.

Lines 135-136: the occurrence frequency estimation, as you describe it here, should refer to all clouds, instead of each type of cloud separately. Please clarify.

Lines 140-142: what are these "brightness temperature biases"? What is the reference value? And what is the extra step to determine the uncertainties that you propose?

Lines 150-151: how did you calculate LTS and EIS? How did you use them in your analysis?

Lines 152-154: Wasn't the occurrence frequency estimated as described in lines 135-136? Please provide a short description of the method mentioned here.

Section 3 general comment: Results are described consistently and are adequately compared with the literature.

Line 163: the cloud categorization appears for the first time in Fig. 3. I suggest to move the categories explanation from the caption of Fig. 3 to a separate table, and present it earlier in the paper.

Lines 188-191: it is a common approach in such cases to use spatial and temporal averaging for a more reasonable comparison. Did the authors try such an approach?

Lines 204-212: Please consider moving this description to Section 2, and adding a table to show this categorization more clearly (see also my comment on line 163).

Line 215: "all types of clouds in SSO are higher…": do you mean that they occur more frequently?

Line 248: The information in Table 1 and Figs. 3b, 4a could also be nicely combined in box plots.

Section 4:

Line 283: In Fig. 5 a LWP threshold of 20 g m$^{-2}$ appears, contrary to the text where a 10 g m$^{-2}$ threshold is mentioned. Please clarify.

Line 292: "… WID is greater than 0.1 m s$^{-1}$ and $V_d$ is greater than 0.0 m s$^{-1}$…". This "and" is an "or" in the diagram of Fig. 5. Please clarify.

Line 307: "By changing … for each range volume". This sentence is not clear, please rephrase.

Line 308: "statistics of the possibility of the cloud phase that may be detected by cloud radar". Please consider replacing with "the possible cloud phase partitioning that may be detected by cloud radar".

Line 347: please replace "e.g." by "i.e.".

Line 358: please replace "least" by "lowest".

Line 385: "mimics" should be replaced by "follows".

Line 419: Is 73% the percentage of all cloud samples available with a threshold of -50 dBZ? Please clarify.

Line 460: please replace "indicating" with "indicative".

Line 483: the term "such as" should be replaced by "i.e.". Also, isn't the LWP for MOL and HOL a full column retrieval? If so, the term "low clouds" should be replaced by "low clouds, including middle and high clouds overlapping cases". Please clarify.

Section 5:

Line 465: please replace "the northern" with "its northern".

Figures:

Figure 1 caption: "Some of the dates have labeled along the shiptracks, which can indicate the direction of the ship traveled". Suggested rephrasing: "Some of the dates are labeled along the shiptracks, indicating the direction of the ship".

Figure 6 caption: please replace "shows" with "is shown" or "appears".

Figure 11 caption: what is the meaning of IOP (also mentioned in line 465)?

---

## Author Comment (AC1)

**Thanks for your suggestions and comments. Please find our point-to-point response in blue color.**

**Response to RC1's Comments:**

Line 1: The title specifies this study as "Cloud properties over the Southern Ocean During the MARCUS field campaign". The "cloud properties" is a broad descriptionincluding macrophysics, microphysics and dynamics, etc. However, the focus of this work is on the investigation of the macrophysics distribution and the cloud phase characteristics. I suggest using a more specific title to be consistent with the scope of the manuscript.

Good suggestion, we have changed out title to 'Cloud phase and macrophysical properties over the Southern Ocean during the MARCUS field campaign'

Line 17: It seems that cloud type is classified with ambiguity. Low-level clouds are identified based on the cloud altitude, while shallow cumulus clouds are classified with their morphology and height. I would prefer to consider the shallow cumulus are a subgroup of low-level clouds instead of a separated group. In the manuscript the authors only use cloud height to identify the shallow/deep cumulus, which is notsufficient. I suggest a thorough revision of these terminologies and use themconsistent in the manuscript to avoid any confusion. Specifically, I suggest using low/middle/high clouds and avoid the "cumulus" category unless a more specific cumulus classification algorithm is being added in the manuscript.

The cloud type classification method was from Xi et al., 2010. Since the ground/ship-based measurements cannot provide cloud 3-D structures, we are trying to classify the clouds only depending on the time series. The original idea of Xi et al. (2010) came from Hogan et al. (2000), and the purpose of the study was to match the temporal measurement (ground-based) with spatial measurement/simulation (Satellite/Model). Therefore, the classification can depend on either height or pressure level. Kennedy et al. (2010) used the same classification method to evaluate the NASA GISS SCM cloud fraction, this method is quite successful in comparing temporal measurement to areal measurements/simulations.

You are right, we did not use a separate algorithm to classify the cumulus, and we modified our manuscript to avoid 'cumulus' for MOL clouds

Xi, B., Dong, X., Minnis, P. and Khaiyer, M. M.: A 10 year climatology of cloud fraction and vertical distribution derived from both surface and GOES observations over the DOE ARM SPG site, J. Geophys. Res. Atmos., doi:10.1029/2009JD012800, 2010

Hogan, R. J., and A. J. Illingworth (2000), Deriving cloud overlap statistics from radar, Q. J. R. Meteorol. Soc.,126, 2903–2909, doi:10.1002/qj.49712656914.

Kennedy, A. D., X. Dong, B. Xi, P. Minnis, A. Del Genio, A. B. Wolf, and M. M. Khaiyer (2010), Evaluation of the NASA GISS single column model simulated clouds using combined surface and satellite observations, J. Clim., doi:10.1175/2010JCLI3353.1

Line 134: Does the cloud occurrence frequency shown in Figure 2 sensitive to the length of the time window? If so, why use 5 min?

The answer is yes. Figure 3 of Xi et al. (2010) has demonstrated how the cloud frequency (FREQ) and amount (amount when present (AWP)) vary with different temporal resolutions (as well as vertical resolutions). Figure 3 shows that although the FREQ and AWP from different temporal

resolutions have similarly shaped vertical distributions, the FREQ values increase and the AWP values decrease as the averaging period increases from 5 min to 6 h. However, the cloud fraction (CF= FREQ * AWP) keeps a constant (the time has to smaller than the lifetime of each type, which will be hard to know unless we have satellite data available to match the areal measurement, like what we did to match the GOES 0.5-hour measurements in Xi et al., 2010).

[Figure]

**Figure 3.** Mean vertical distributions of (a) FREQ, (b) AWP, and (c) CF derived from the ARM radar-lidar pair observations with a vertical resolution of 90 m and temporal resolutions of 5 min, 1 h, 3 h, and 6 h at the ARM SGP site, 1997–2006.

Line 188 also mentions the relative low cloud fractions compared to Satellite results is due to the inconsistent resolution. Adapting the wind speed from sounding, a reasonable "equivalent temporal resolution" of the surface-based radar may be obtained. For example, considering the windspeed as 10m/s, a 2-grid box(~200km) corresponding to approximately 5.5hours of the radar observation (not considering the speed of ship). Thus, does the cloud frequency estimated from a longer time window is closer to the spaceborne results? I'm not suggesting performing a sensitive test for Fig.2, but just want to remind the authors that it is necessary to discuss this issue furthermore for the presented results to be better used for model validation, as the authors mentioned inthe introduction.

Yes, we added more discussions in the revision regarding to Figure 2 based on the conclusion in Xi et al. (2010).

Figure 1 of Xi et al. (2010) has illustrated the temporal-spatial CF comparison between ARM SGP ground-based and GOES satellite. We concluded that the 0.5-hr averaged ARM CFs agreed well with 0.5° GOES observations, while 4-hr averaged ARM CFs matched well with 2° GOES results.

[Figure]

**Figure 1.** Dependence of (a) cloud frequency of occurrence (FREQ), (b) amount when present (AWP), and (c) cloud fraction (CF) on temporal resolutions of Atmospheric Radiation Measurement Program (ARM) surface radar–lidar observations during the period 1997–2006 and on grid boxes of satellite observations during the period from May 1998 to December 2006 at the ARM Southern Great Plains (SGP) site.

Line 200: How close? What is the specific CF from Listowski et al, 2019?
This statement is based on the comparison of the color bars of Figures 7 & 9 in Listowski et al. (2019). The low clouds (below 3 km) are dominated in SON and DJF in this study compared to the Listowski et al. (2019) defined Weddell Sea sector (WSS) and Amundsen–Ross sector (ARS regions) in the SO, especially the ARS region is the MARCUS experiment domain.

Line 214: I also mentioned in the previous comment. The classified cloud type is solely based on cloud heigh, it is not clear why the MOL is related to shallow cumulus.
Changed 'shallow cumulus' to MOL in this revision.

Line 215: Are you refereeing to occurrence frequency is higher?
Yes, we rephrased the sentence for clarifying. Thanks.

Line 226: For Figure 3. Why use the deepest cloud thickness instead of the mean/median thickness? It is hard to interpret the thickness distribution in a statistical perspective as the deepest thickness may be the results of two "outliers" of the cloud in each group. Besides, I suggest using the box plot to present the results shown in Fig. 3b.
Thanks for the suggestion. The purpose of Figure 3b is to show where is mean cloud boundary for

each selected category. The bars in Figure 3b are the mean values listed in Table 2, that is, the mean thickness for each category except for HOL. The sentence is changed in the revision as: Figure 3b shows the vertical locations of different types of cloud layers, which represent the mean $H_{top}$ and $H_{base}$ listed in Table 2 for any type of cloud.

Line 232: How the stability is estimated and what is the value for NSO?
The stability is estimated by the LTS calculated from the potential temperature difference between the surface and 700 hPa based on the ERA-Interim reanalysis during the time periods when the low-level clouds appeared along the ship tracks. The boundary layer over NSO is relatively more stable than over SSO based on lower troposphere stability (LTS) analysis (12.2-15.32 K over NSO vs. 11.48-13.29 K over SSO).

Line 235: …HGH and HOM peak "are" less than 10 g m$^{-2}$
Thanks, it is changed.

Line 241: Same as before, the shallow convective clouds are not clearly identified.
You are right, we removed the 'shallow convective' MOL clouds.

Line 248: Table1: Compared to the mean LWP, the std (uncertainty) is too large and indicates the result is statistically insignificant. This may be due to the insufficient MWR observations being collected in the analysis. Have you considered the period when MWR is wet and thus the LWP retrieval becomes unreliable? The maximum LWP value (1937 g m$^{-2}$) seems too large to be accurately retrieved by MWR.
The max values of LWP from statistical method are much greater than what we used. We understand the wet dome may cause bias, and thus performed a sensitivity study in Tian et al., (2019) for shallow and deep convective clouds and concluded that the MWR retrieved cloud LWPs are reasonable as long as the rain rate < 2 mm/hr, that is, the wet dome is insignificant for Rain Rate <2 mm/hr.

Tian, J., X. Dong, B. Xi, C.R. Williams, and P. Wu, 2019: Estimation of Liquid Water Path in Stratiform Precipitation Systems using Radar Measurements. Atmos. Meas. Tech. 12, 3743–3759, https://doi.org/10.5194/amt-2018-388.

Line 250: Does the "non-contiguous clouds" refers to multi-layer clouds?
Yes.

Line 251-252: Please add the corresponding statistics in Table 1 to support this statement.
In column 7 of Table 2, we include both single- and multi-layered statistics for the HML clouds.

Line 266: Same as before, I suggest using the mean cloud thickness instead of the "maximum" cloud thickness before making further statements.
We have explained the reasons to use the max cloud top and min cloud base in last question.

Line 288:290: What is the meaning of the "majority of WID?" Why not using the mean/median WID? What is the radar temporal resolution?
We did use in-site measurements to calculate the mean WID while ice particles present, which

value is 0.52 m/s. The reason using 'majority of WID' is that we have to determine the dominant phase in each 5-minute. Since not all the measurements have 10-s data, we have to process all the measurements into 5-minute resolution.

Line 292: And or "or"?
Changed to 'or', thanks for catching our mistake.

Line 289L Any supporting information regarding on the chosen of 0.4m/s of WID?
When the 2DS detected the ice particles, the WID from in-situ radar usually are >0.4 m/s.

Line 288-Line 294: Please rephrase the description of the classification algorithm, which is not consistent with Figure 5.
We revised Figure 5.

Line 293: LWP threshold (10 gm$^{-2}$) from the manuscript and Figure 5 (20 g m$^{-2}$) is not consistent.
Thanks for catching our mistake, the figure 5 has changed to 10 gm$^{-2}$.

Line 305-310: Not quite understand why this analysis is helpful to validate the proposed classification algorithm.
We are trying to integrate all 1-s in-situ measurements into different temporal resolutions to mimic the radar observations. Since the in-situ cloud microphysical measurements can tell us the phase of the cloud, it allows us to see the percentage variations of cloud phase, by changing integration time of in-situ sampling to mimic what the radar may observe the cloud for each range volume.

Line 315: I cannot tell the backscatter difference for Fig.6e, please adjust the color bar range to make the figure more distinguishable. Also, the backscatter unit is missing.
We add the unit and replot it.

Line 321-323: From Fig. 6c, the identified ice period from radar around 22 UTC corresponding to low LDR value in Fig 6.d, which seems contradictory to this statement.
In the revised plot, Fig. 6d clearly shows LDR > 0.1 for both mixed-phase and ice clouds.

Line 337-339: It is necessary to present the SOCRATES results if this can further support the classification output. One thing should be considered for the onboard radar is that the Doppler spectrum width may be much higher than the surface-based radar as for the enhanced Doppler spectrum broadening caused by the moving of aircraft.
Thanks for the suggestion, we did include the SOCRATES analysis in Table 2b. We talked the Doppler effect in our ATMO physics classes and understood that it is relative to movement in the same direction. We also did some research to use ARM ground-based observed Doppler velocity and spectrum for retrieving cloud microphysical properties (Fig. 3 of Tian et al., 2019; Wu et al., 2020). In this study, the onboard radar sends the signals (assuming the time of transmitted and received signals is short enough comparing to aircraft speed) in the perpendicular to the movement of the aircraft, that is, there is no relative movement between radar signals and clouds. So we are not sure how we can quantitatively estimate the Doppler broadening in this study. Please let us know if you have more suggestions on the order of magnitude. Thanks.

Line 345-346: What is the difference between mixed-phase and "other" clouds?

'Other' is also mixed-phase cloud but have different vertical distribution of liquids in it, compared to the 'mix-phase' cloud. We think it will be very interesting to include it in a stand-alone group. We will give a clearer definition about "Other" in the revision.

Line 364: Drizzle is one kind of the liquid water droplets; why drizzle occurs in ice cloud?

From radar reflectivity, we can only tell there are effective radar reflectivity below the cloud base (defined by ceilometer/MPL), we account these actively falling particles as 'drizzle', which could be ice crystals. We revise it in the manuscript to clarify that. Thanks for your suggestion.

Line 411-Line 413: This comparison only make scene if the cloud phase classification algorithm is same for these two studies.

Yes, the cloud phase classification algorithm is almost identical. The algorithm used in this study is very similar to the method of Shupe et al. (2005) which was developed based ARM NSA datasets. The comparison between this study with Shupe's one is listed in Table 2a.

The vertical distributions of the radar reflectivity over ARM NSA site could be quite different from those during MARCUS. The most outstanding characteristic of Arctic mixed-phase clouds (also low clouds) is featured with the liquid-topped cloud layer with ice cloud layer beneath (Qiu et al. 2015). When we looked day-by-day radar image and cannot find any long-lasting liquid-topped mixed-phase cloud layers during MARCUS. This is the largest difference between two regions.

Qiu, S., X. Dong, B. Xi. J. Li, 2015: Characterizing Arctic mixed-phase cloud structure and its relationship with humidity and temperature inversion using ARM NSA Observations. JGR, 120, 7737-7746, doi:10.1002/2014JD023022.

Line 417: Radar reflectivity need to be calibrated before used for analysis. (Mace., 2021)

We have personally contacted the ASR data management team regarding that. They are aware of the radar reflectivity during MARCUS has low bias but don't know exactly how much lower. We just try our best to get an estimation here.

Line 429: This statement is valid only if the vertical air velocity is negligible compared to the terminal velocity of cloud/ice particles.

Yes, you are right, it is hard to get vertical air velocity when the particle moves the same direction as air. However, drizzle forms frequently in liquid clouds, and the drizzle falls actively downward, which enhances downdraft. In our understanding, the vertical velocity of cloud droplets (r< 20 um) is similar to air vertical motion, while for large drizzle drops and ice crystals (> 100 um) their terminal fall velocity should be much larger than air motion. In this situation, we can treat the radar Doppler velocity as drizzle or ice crystal terminal fall velocity with small uncertainty. We will include this discussion in the revision to clarify this point.

---

## Author Comment (AC2)

**Thanks for your suggestions and comments. Please find our point-to-point response below in blue color.**

**Response to RC2's Comments:**

General comment:
This study presents a statistical analysis of cloud types, their occurrence and macrophysical properties over a Southern Ocean region (and separately its northern and southern parts), based on measurements from the MARCUS campaign. In most cases, presentation of results, discussion and comparison with other studies is adequate. The results are relevant for AMT and can be useful for evaluation studies of satellite retrievals and model simulations. For these reasons, I recommend that this manuscript is accepted for publication after major revisions. These regard mainly structural issues (particularly in Section 2) and several points throughout the text that need to be clarified.

Specific comments:

Section 1:
Lines 97-98: "which in-situ … this analysis". This part of the sentence is not clear. Do you mean that in-situ measurements from SOCRATES were used here as reference? Please clarify and rephrase.
Yes, we are using SOCRATES measurements as a reference because these two experiments do not have overlapped in the same location but have the overlapped period (Jan-Feb. 2018). The sentence is rephrased to 'In this study, the aircraft in-situ measurements from SOCRATES are used as the reference for the analysis.'

Lines 99-102: while the objectives of the MARCUS campaign are clearly stated in this sentence, it is not clear what the focus of this study is. While it is described in the next paragraph, this sentence needs rephrasing.
We changed the entire sentence to: 'Our study will focus on cloud macrophysical properties and cloud phases along the ship tracks.'

Section 2 general comment: It was difficult for me to follow the text in Section 2. The authors describe data sets, algorithms and methods in an unclear and mixed way. Please provide a more concise description of data sets and methods by considering the following suggestions:
- It would be helpful for the reader if you provide a table with all the cloud parameters measured and analyzed in this study.
- For each cloud parameter, please provide a short description of the instrument and method used for its measurement or retrieval.

We have rearranged the texts in Section 2 to provide better readability and provided a table with a brief description for each instrument, its measurement and uncertainty. More detailed information can be found from Mace et al. (2021).

Mace, G. G., Protat, A., Humphries, R. S., Alexander, S. P., McRobert, I. M., Ward, J., Selleck, P., Keywood, M. and McFarquhar, G. M.: Southern Ocean Cloud Properties Derived From CAPRICORN and MARCUS Data, J. Geophys. Res. Atmos., doi:10.1029/2020JD033368, 2021.

Line 118: please provide a reference or short description for the AMF2 instruments suite.
The following references are cited in Section 2.

McFarquhar, G., Bretherton, C., Alexander, S., DeMott, P., Marchand, R., Protat, A., Quinn, P., Siems, S., Weller, R., Wood, R.: Measurements of Aerosols, Radiation, and Clouds over Sothern Ocean (MARCUS) Science Plan, DOE ARM Climate Research Facility., DOE/SC-ARM-16-011, available at: http://arm.gov/publications/programdocs/doe-sc-arm-16-011.pdf, 2016.

McFarquhar and coathors, 2021: Observations of Clouds, Aerosols, Precipitation, and Surface Radiation over the Southern Ocean: An Overview of CAPRICORN, MARCUS, MICRE, and SOCRATES, BAMS, https://doi.org/10.1175/BAMS-D-20-0132.2.

Lines 135-136: the occurrence frequency estimation, as you describe it here, should refer to all clouds, instead of each type of cloud separately. Please clarify.
For each 5-min sample, we can only determine one type of cloud (low, mid, high, etc) and thus the column cloud fraction of that classified type of cloud within 5 min. Thus, we can further estimate the occurrence frequency of each type of cloud separately. We have also clarified this statement in the revised manuscript.

Lines 140-142: what are these "brightness temperature biases"? What is the reference value? And what is the extra step to determine the uncertainties that you propose?
Since the retrieved LWP and PWV are based on the MWR measured brightness temperatures at two frequencies, any biases on the brightness temperatures will affect these retrievals. Therefore, we propose an extra step to determine the LWP uncertainty during MARCUS. Based on the sounding temperature profiles, we can identify clouds that are not likely to contain liquid (e.g., pure ice-cloud), then we can estimate the LWP uncertainty based on their corresponding retrieved LWP values. From the PDF analysis, the LWP uncertainty is estimated as 10 g m$^{-2}$ during MARCUS IOP.

Lines 150-151: how did you calculate LTS and EIS? How did you use them in your analysis?
The lower tropospheric stability (LTS) is calculated from the potential temperature difference between the surface and 700 hPa based on the ERA-Interim reanalysis to assess the boundary-layer stabilities when the low-level clouds appeared along the ship tracks. We did not use EIS so it was removed.

The following statements are in Section 3:
By analyzing the ERA-Interim reanalysis (not shown), the 850 hPa geopotential heights show persistent westerlies with slightly higher geopotential heights over the northwest corner of the domain, which may closely relate to the higher Htop over NSO than over SSO. Furthermore, the boundary layer over NSO is relatively more stable than over SSO based on lower troposphere stability (LTS) analysis (12.2-15.32 K over NSO vs. 11.48-13.29 K over SSO).

Lines 152-154: Wasn't the occurrence frequency estimated as described in lines 135-136? Please provide a short description of the method mentioned here.
The radar records measurement every 2-second, which shows the part of the column that is cloudy, so the column cloud fraction can be given by the total cloudy samples divided by 150 samples (assuming all the samples within 5 minutes have valid measurement). The statement is rephrased accordingly.

Section 3 general comment: Results are described consistently and are adequately compared with the literature.)
Thanks.

Line 163: the cloud categorization appears for the first time in Fig. 3. I suggest to move the categories explanation from the caption of Fig. 3 to a separate table, and present it earlier in the paper.
Please refer to Table1 and Figure 6 in Xi et al., 2010. The detailed description is also added in Section 2.

Lines 188-191: it is a common approach in such cases to use spatial and temporal averaging for a more reasonable comparison. Did the authors try such an approach?
Because CloudSat has difficulty retrieving the clouds below ~1 km, we cannot directly compare these two results via temporal and spatial averaging. We did successfully match the GOES and ground-based measurements in Xi et al., 2010.

In fact, the reviewer 1 has raised the same question. Here is what we response. Figure 1 of Xi et al. (2010) has illustrated the temporal-spatial CF comparison between ARM SGP ground-based and GOES satellite observations. We concluded that the 0.5-hr averaged ARM CFs agreed well with 0.5° GOES observations, while 4-hr averaged ARM CFs matched well with 2° GOES results.

[Figure]

**Figure 1.** Dependence of (a) cloud frequency of occurrence (FREQ), (b) amount when present (AWP), and (c) cloud fraction (CF) on temporal resolutions of Atmospheric Radiation Measurement Program (ARM) surface radar–lidar observations during the period 1997–2006 and on grid boxes of satellite observations during the period from May 1998 to December 2006 at the ARM Southern Great Plains (SGP) site.

Lines 204-212: Please consider moving this description to Section 2, and adding a table to show this categorization more clearly (see also my comment on line 163).
This description is moved to Section 2.

Line 215: "all types of clouds in SSO are higher…": do you mean that they occur more frequently?

Yes, we have rephrased the sentence as 'all types of clouds in SSO have higher frequency of occurrence than those in NSO except HOL.'

Line 248: The information in Table 1 and Figs. 3b, 4a could also be nicely combined in box plots.

We think the mean and standard derivation on top of each bar might be too difficult to read, so we think it is better to keep table 1.

Section 4:

Line 283: In Fig. 5 a LWP threshold of 20 g m$^{-2}$ appears, contrary to the text where a 10 g m$^{-2}$ threshold is mentioned. Please clarify.

Thanks for catching our mistake. It has changed to 10 g m$^{-2}$.

Line 292: "… WID is greater than 0.4 m s$^{-1}$ and V$_{d}$ is greater than 0.0 m s$^{-1}$…". This "and" is an "or" in the diagram of Fig. 5. Please clarify.

Thanks for catching our mistake. It has changed.

Line 307: "By changing … for each range volume". This sentence is not clear, please rephrase.

Change to 'integration time'.

Line 308: "statistics of the possibility of the cloud phase that may be detected by cloud radar". Please consider replacing with "the possible cloud phase partitioning that may be detected by cloud radar".

Changed, thanks for the suggestion.

Line 347: please replace "e.g." by "i.e.".

Changed.

Line 358: please replace "least" by "lowest".

Changed.

Line 385: "mimics" should be replaced by "follows".

Changed.

Line 419: Is 73% the percentage of all cloud samples available with a threshold of -50 dBZ? Please clarify.

No, the denominator was always all the measurements. So, by changing the threshold from -40 dBZ to -50 dBZ, we included 17.4% more data. We added the following sentence for clarification "If we used the threshold of -50 dBZ, then we would have 90.4 % cloud samples, which gained 17.4% more samples on top of the -40 dBZ threshold."

Line 460: please replace "indicating" with "indicative".

Changed, thanks.

Line 483: the term "such as" should be replaced by "i.e.". Also, isn't the LWP for MOL and HOL a full column retrieval? If so, the term "low clouds" should be replaced by "low clouds, including middle and high clouds overlapping cases". Please clarify.

Rephased to 'The mean *LWP*s for LOW, MID and HOL clouds over NSO, range from ~130 to 150 g m$^{-2}$, while the mean *LWP*s (~270 g m$^{-2}$) for MOL and deep convective clouds (HML), are

much higher than the same types of clouds over SSO'

Section 5:
Line 465: please replace "the northern" with "its northern".
Changed.

Figures:
Figure 1 caption: "Some of the dates have labeled along the shiptracks, which can indicate the direction of the ship traveled". Suggested rephrasing: "Some of the dates are labeled along the shiptracks, indicating the direction of the ship".
Changed, thanks the suggestion.

Figure 6 caption: please replace "shows" with "is shown" or "appears".
Changed.

Figure 11 caption: what is the meaning of IOP (also mentioned in line 465)?
Changed to 'Intensive observational period (IOP)'.

---

## Author Response (AR2)

**Thanks for your suggestions and comments. Please find our point-to-point response in blue color.**

The authors implemented appropriate changes, based on the previous review round. Several points were clarified, and the manuscript has overall improved. I recommend that it is accepted for publication. Before the final submission, I would like to ask the authors to double-check the following two points (line numbers refer to my previous review comments):

1. Lines 135-137: The occurrence frequency estimation is clarified in the authors' reply. In the manuscript, however, the original description is just deleted. I could not find this clarification. Please check.

**Thanks for catching the missing definition. It will be more clearly to add "The cloud occurrence frequency can be determined through two steps: the column cloud fraction is simply the ratio of cloudy samples to the total observations in every 5-min; the occurrence frequency for each type of cloud during the entire time period equals the ratio of the number where column cloud fraction is greater than zero to the total 5-min samples." In lines 155-158**

2. Lines 188-191: Again, I could not find in the manuscript the relevant revised discussion that clarifies the issue. Please check and revise if needed.

**The discussion is at lines 229-236.**

---

## Author Response (AR3)

**Dear Editor,**

**Thanks for accepting our manuscript. Please find our point-to-point response (in blue color) to the Remarks from the preceding review file validation.**

Please ensure that the colour schemes used in your maps and charts allow readers with colour vision deficiencies to correctly interpret your findings. Please check your figures using the Coblis – Color Blindness Simulator (https://www.color-blindness.com/coblis-color-blindness-simulator/) and revise the colour schemes accordingly. With the next revision, please number your tables consecutively: Table 1,Table 2, Table 3 (a,b) (not Table 3A, Table 3B), Table 4, etc. More see: https://www.atmospheric-measurement-techniques.net/submission.html#figurestables

**We re-plot some figures using the same data and check them via the Color Blindness Simulator to ensure they can be correctly interpreted. The corresponding figure captions are also revised.**

**All the table numbers are correctly labeled in the revised manuscript.**